# Electrochemical growth mechanism of nanoporous platinum layers

Sarmiza-Elena Stanca [1✉], Oliver Vogt[2], Gabriel Zieger[1], Andreas Ihring[1], Jan Dellith[1], Andreas Undisz[3], Markus Rettenmayr[4] & Heidemarie Schmidt [1,5✉]

Porous platinum is a frequently used catalyst material in electrosynthesis and a robust broadband absorber in thermoelectrics. Pore size distribution and localization determine its properties by a large extent. However, the pore formation mechanism during the growth of the material remains unclear. In this work we elucidate the mechanism underlying electrochemical growth of nanoporous platinum layers and its control by ionic concentration and current density during electrolysis. The electrode kinetics and reduction steps of $PtCl_4$ on platinum electrodes are investigated by cyclic voltammetry and impedance measurements. Cyclic voltammograms show three reduction steps: two steps relate to the platinum cation reduction, and one step relates to the hydrogen reduction. Hydrogen is not involved in the reduction of $PtCl_4$, however it enables the formation of nanopores in the layers. These findings contribute to the understanding of electrochemical growth of nanoporous platinum layers in isopropanol with thickness of 100 nm to 500 nm.

[1] Leibniz Institute of Photonic Technology, Jena, Germany. [2] Deutsche METROHM GmbH & Co. KG, Filderstadt, Germany. [3] Institute of Materials Science and Engineering (IWW), Technische Universität Chemnitz, Chemnitz, Germany. [4] Otto-Schott-Institute of Material Research, Friedrich-Schiller-Universität Jena, Jena, Germany. [5] Institute for Solid State Physics, Friedrich-Schiller Universität Jena, Jena, Germany. ✉email: sarmiza.stanca@leibniz-ipht.de; heidemarie.schmidt@leibniz-ipht.de

The scientific and technological interest in porous platinum is triggered by its large broadband absorption and small reflectance in the region from ultraviolet to infra-red, by its chemical inertness in air and water, and by its mechanical and thermal stability. Already more than 126 years ago porous platinum could be deposited on platinum cathodes in aqueous electrolytes of chloroplatinic acid by means of aqueous electrochemistry with the addition of copper or lead salts[1]. Two years later, Kohlrausch platinized platinum electrodes[2] at high current densities from an aqueous solution of $PtCl_4$ with the addition of lead acetate and suggested that it was a two-step process in which hydrogen was first formed and subsequently reduced the platinic chloride[2]. Moreover, under extreme conditions of a platinized electrode that had been maintained in the deoxygenated 1 M HCl for 2 days, a four-electron step reduction of the aqueous electrolyte of chloroplatinic acid $H_2[PtCl_6]$ was observed[3]. The electrodeposition of porous platinum layers in aqueous media on various cathodes was laboriously studied[3–6]. Feltham and Spiro[3] mainly contributed to the understanding of the nucleation and growth mechanism of the platinum layer in aqueous media in 1972. Recently, we demonstrated the electrochemical growth of porous platinum layers from a non-aqueous solution of $PtCl_4$[7]. For example in microtechnology, electrochemistry from non-aqueous solutions is preferred over electrochemistry from aqueous solutions.

In this work, we study the electrochemistry of platinum layers in aqueous and non-aqueous media and reveal the mechanism underlying the observed formation of micro- and nano-sized porous platinum layers on the electrodes (Fig. 1) with different surface structures. The interpretation and understanding of the results are an important prerequisite for controlling the growth of the nanoporous layers in terms of thickness, structure and porosity (Figs. 2–5). We used two complementary methods, staircase and linear sweep cyclic voltammetry (CV)[8–11]

and electrochemical impedance spectroscopy (EIS)[12–15], to study electrochemistry of nanoporous platinum from aqueous and from non-aqueous solutions of $PtCl_4$ without and with additives. High-resolution transmission electron microscopy (HRTEM), scanning electron microscopy (SEM), X-ray diffraction (XRD) and energy-dispersive X-ray (EDX) diagrams recorded at different stages of the electrodeposition provide information on the layer morphology and crystallinity (Figs. 1–5). The electrochemical study reveals a two-step reduction of platinum (IV) during electrodeposition in both aqueous and non-aqueous medium (Figs. 6–8a, b). The process occurs independently of the hydrogen gas formation, which causes the porosity development of the platinum layer and not the platinum ion reduction. By increasing the rate of the reaction and shifting the oxido-reduction potential to more positive values, one could involve additives in the formation of a porous platinum layer. The staircase CV wave of oxidation in the interval of $0.8 \div -0.5$ V, which is detectable at a scan rate of 0.25 V/s (Fig. 6f) and which is attributed to the platinum oxidation, is no longer observable in the system containing additives (Fig. 6g). That implies that it has undertaken an irreversible reduction or that the additive alters the stability of the oxidised platinum species, thus shifting the oxidation potential to higher values. Meanwhile, the impedance measurements (Figs. 8c, d and 9) reveal two kinetic processes at the applied potential in the interval from 0 to $-0.6$ V and three kinetic processes at the applied potential in the interval from $-0.7$ to $-1$ V. Presented results further elucidate the mechanism underlying electrochemical growth of nanoporous platinum layers.

## Results

We present here results on the electrochemistry of adherent porous platinum layers of controlled thickness and porosity in aqueous and non-aqueous solutions of $PtCl_4$ without additive and

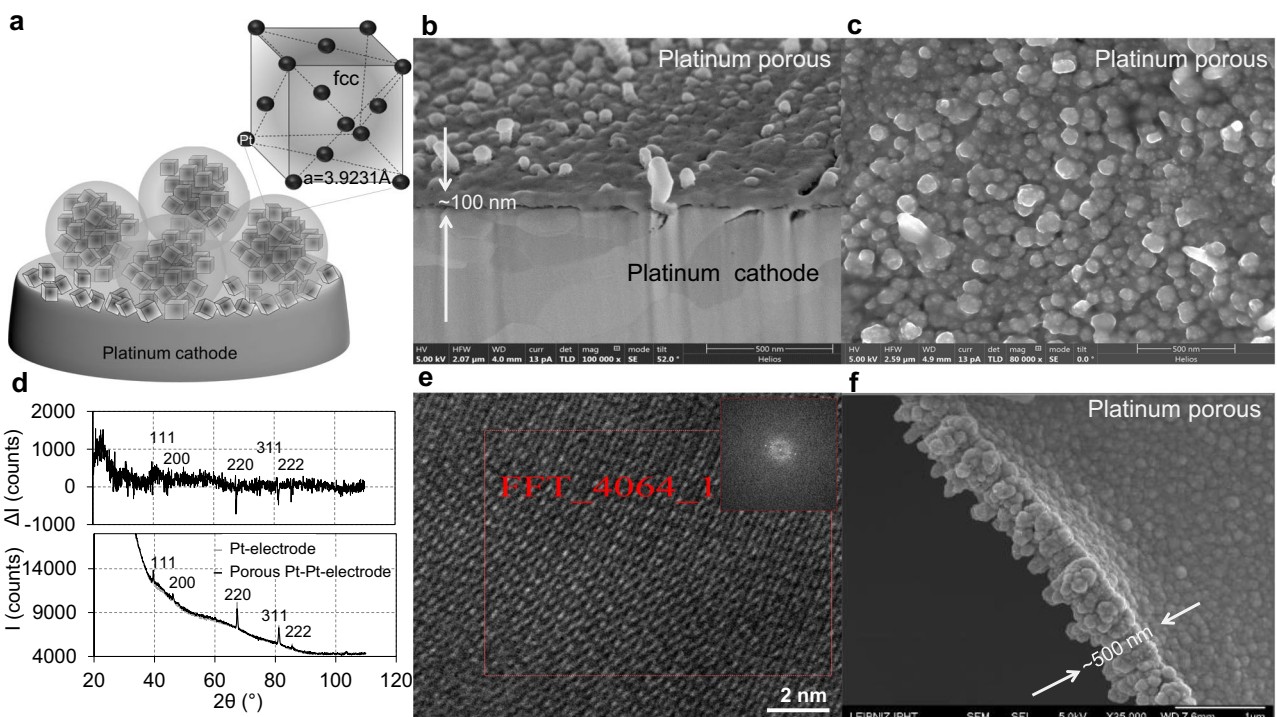

**Fig. 1 Porous platinum layer electrochemically grown on platinum electrodes. a** Scheme of platinum crystal assembly on the platinum cathode; not to scale; **b, c** SEM of the porous platinum layer electrochemically grown in 90 s on the platinum cathode at $-1$ V using the reduction of 0.5% $PtCl_4$, 0.01% Pb $(CH_3COO)_2$ in isopropanol (g/g); **d** XRD diagram and the difference XRD spectra of porous platinum film growth on platinum and platinum cathode; **e** HRTEM images of the porous platinum growth on a platinum-coated copper grid. The Fast Fourier Transform of the indicated area; **f** SEM of the porous platinum layer grown on a platinum-coated copper grid similar to the one investigated in (**e**).

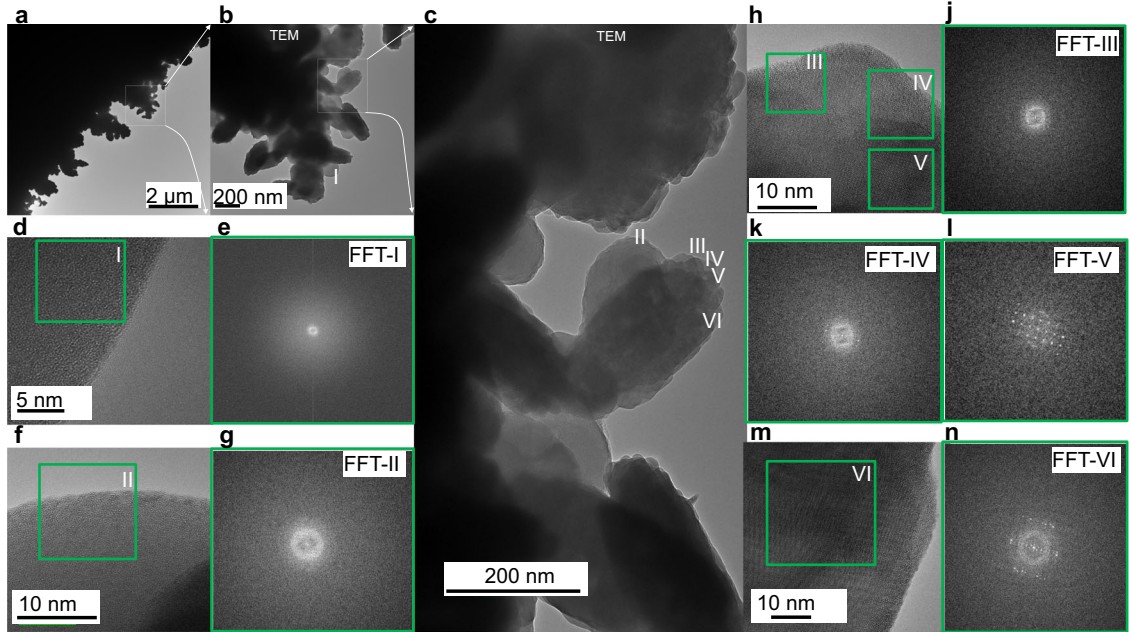

**Fig. 2 Porous platinum layer seen in transmission electron microscopy. a–n**, TEM images of the porous platinum growth on a platinum-coated copper grid (**a–d**, **f**, **h**, **m**) and the Fast Fourier Transform, FFT, (**e**, **g**, **j**, **k**, **l**, **n**) of each indicated areas (I–VI).

in the presence of the different additives. Kurlbaum and Lummer used copper sulphate and lead acetate to increase the adherence of porous platinum on the substrate; they obtained better results using lead acetate[1]. Feltham and Spiro[3] also have shown the adherence of platinum black in the presence of lead acetate. We have also tested copper acetate, copper sulphate, acetic acid and lead acetate as additives in the aqueous and non-aqueous electrochemical baths and observed that the adherence of the porous platinum layer on different substrates (copper, platinum, aluminium, indium–tin, silver and gold) is best supported if lead acetate is used as an additive.

**Cyclic voltammetry**. An electrochemical study of the reduction process of an aqueous and non-aqueous PtCl$_4$ solution was performed in a closed microcell (see "Methods") equipped with four platinum electrodes, each of which had a diameter of 250 μm (Fig. 6a–e). The exchange current density at the interface between electrode and solution is a direct measure of the electrode's readiness to proceed with electrochemical reaction[8–10]. Rapid and qualitative conclusions on the electrode-solution kinetics can be drawn from the exchange current density recorded as a function of applied potential in cyclic voltammograms. Observed peaks in recorded cyclic voltammograms are interpreted as sweep ranges where kinetics at the electrodes are determined by both charge transfer and mass transfer (chemical reaction), whereas the observed "S" shaped curves in recorded cyclic voltammograms are interpreted as sweep ranges where kinetics at the electrodes is determined by charge transfer processes only[8–10]. Because in this work we study the chemical reaction processes during electrochemistry of nanoporous platinum layers formation, we mainly focus on the analysis of peaks in recorded CVs. The CVs recorded for PtCl$_4$ in isopropanol of concentrations 0.05%, 0.1%, 0.2%, 0.3% and 0.4% (g/g) exhibit undefined peaks, therefore, for further experiments we used PtCl$_4$ of higher concentrations (Figs. 6 and 7). The evolution of the stair-case and linear sweep cyclic voltammograms of 0.5% PtCl$_4$ in isopropanol in the presence/absence of additive (0.01% Pb(CH$_3$COO)$_2$) was recorded between +2 and −2 V at 0.01, 0.1 and 1 V/s and plotted as shown in Fig. 7b. Furthermore, the CVs of PtCl$_4$ (0.5, 1 and 2% g/g)

in isopropanol at 0.1 V/s were compared to the CV of pure isopropanol, which is also a preferred solvent for its large electrochemical window[11] (black line) (Fig. 7a). For comparison, the CVs of PtCl$_4$ in the aqueous solution was also recorded (Fig. 8a, b). For the same concentration of PtCl$_4$ (0.5% g/g) the current intensities increase from $1.5 \times 10^{-6}$ A (non-aqueous) (Fig. 6f) to $1.5 \times 10^{-5}$ A (aqueous) (Fig. 8a) showing a faster reaction rate in aqueous versus non-aqueous medium.

**Electrochemical impedance spectroscopy**. This method relies on the polarisation of the electrochemical cell at a fixed voltage followed by a perturbation to the system. The relative response amplitude and phase shift between input and output voltage signals are recorded. The output voltage changes as a function of applied frequencies when different frequencies can separate processes with different kinetics. A capacitance-controlled response shows that the chemical process is slower than the rate with which the applied field is changed. Instead, a diffusion-controlled response shows a reverse event.

EIS can be used to analyse the porous platinum layer formation (growth rate, thickness) in dependence on the applied voltage difference between the electrodes. This is useful for optimising and controlling the electrochemical deposition of porous platinum.

We recorded EIS spectra during electrochemical deposition of porous platinum layers for PtCl$_4$ 0.5% (g/g) in an aqueous medium (Fig. 8c, d) and in isopropanol (Figs. 9 and 10) at different potentials (0, −0.5, −0.6, −0.7, −0.8, −0.9, −1, −1.1, −1.2, −1.3, −1.4 and −1.5 V) with 10 mV test amplitude in the frequency range of 0.1 ÷ 10$^5$ Hz, starting from the highest frequency. Nyquist plots show the kinetic change from one reaction process (at 0 V, −0.5 V) to two reaction processes (−0.6 V ÷ −1 V) and provide evidence of the stability of the formed porous platinum layer. The effect of the additive on the formation of porous Pt in isopropanol is revealed by the comparison of the corresponding Nyquist diagrams in Fig. 9. Detailed analysis of the impedance plot is challenging because the thickness of the nanoporous Pt layer increases in the frequency range of 0.1 ÷ 10$^5$ Hz. We performed the experiments by changing only one

parameter between the systems. We gradually recorded the impedance plots for 0.5% $PtCl_4$ in isopropanol in the absence (Fig. 9a) and presence (Fig. 9b) of the additive. We follow the observable changes in impedance curves as functions of the applied potential, time and frequencies (Figs. 9c–f and 10) and correlate these changes with observed structural changes in SEM images and literature data[12–15]. The apparition of the second loop in the impedance plot is connected with the layer detachment[13]. This is the case of the $PtCl_4$-isopropanol system without additive at a potential more negative than −0.6 V and at small test frequencies (Fig. 9a). The absence of the second loop and the presence of the capacitive tails (charge transfer) indicate that the newly formed layer of platinum is attached to the electrode (Fig. 9b) and behaves as a new compact electrode. Between the first semicircle and the capacitive tails, we observe a "resistive" interval: a shorter segment at −0.7 V and a larger segment at −1 V (Figs. 9 and 10b). This constant increase in $Z_{re}$ can be attributed to the adherent layer growth as indicated by the SEM. In contrast, without the additives at −0.7 V ÷ −1 V, the apparition of the second loop indicates non-adherent layer growth. Moreover, Fig. 9c–f shows that the $Z_{re}$ values recorded at −0.7 V ÷ −1 V without additive (Fig. 9c) are smaller in comparison to the $Z_{re}$ values recorded at −0.7 V ÷ −1V with additives (Fig. 9d) after 150 s. This difference occurs because of insufficient adherence of the layer prepared in the absence of additive. As a consequence, the electrode remains void of layers and displays higher electrical conductivity and lower impedance. $Z_{im}$ constantly increases with the more negative applied potential at times shorter than 130 s (Fig. 9e). This capacitive increase is slightly more accentuated in the presence of additives (Fig. 9f). At longer times than 150 s, $Z_{im}$ decreases in both cases. One observes that in the presence of additives $Z_{im}$ is more increased at low frequencies and $Z_{re}$ is more increased at higher frequencies. This

is not unexpected because the imaginary part (capacitive part) dominates at lower frequencies and because the real part (resistive part) dominates at larger frequencies. The impedance measurements (Figs. 9–10) show the following behaviour in the frequency range from 0.1 to $10^5$ Hz:

1. an increased electrical resistance of $5 ÷ 8 × 10^5 \, \Omega$ through the adherent porous platinum layer formation, which is correlated with two connected semicircles of the impedance plot; the second one is at low capacitance with a steep capacitance tail of almost 90°;

2. a relatively lower electrical resistance of $1.5 ÷ 3.5 × 10^5 \, \Omega$ during the non-adherent platinum layer formation, which is correlated with two semicircles near each other without a diffusion tail; instead, the second semicircle becomes an incomplete circle (inductive loops often met at corrosion).

**Morphology and light trapping.** Electrolysis medium considerably influences the platinum layer structure and morphology due to the difference in hydrogen formation (more abundant in aqueous media) and $Pt^{4+}$ reduction reaction rate (lower in non-aqueous medium). Using SEM, the porous platinum that was electrochemically formed on platinum electrodes in nonaqueous media (Figs. 1, 3 and 4) and aqueous media (Fig. 5) was examined. The electron micrographs show a porous layer with a thickness of 100 nm (Fig. 1b) and 500 nm (Fig. 1f). Backscattering electron micrograph (Fig. 3d, e) indicates the fine structures of the platinum porous layer, that nanopores between the platinum crystals are formed. The crystallinity was confirmed by XRD (Fig. 1d) and Fast Fourier Transform (FFT) image analysis of HRTEM recorded at the margins of the porous layer growth (Figs. 1e and 2) on a platinum-coated copper grid. In the range larger than 500 nm the assemblies exhibit various shapes some of them are

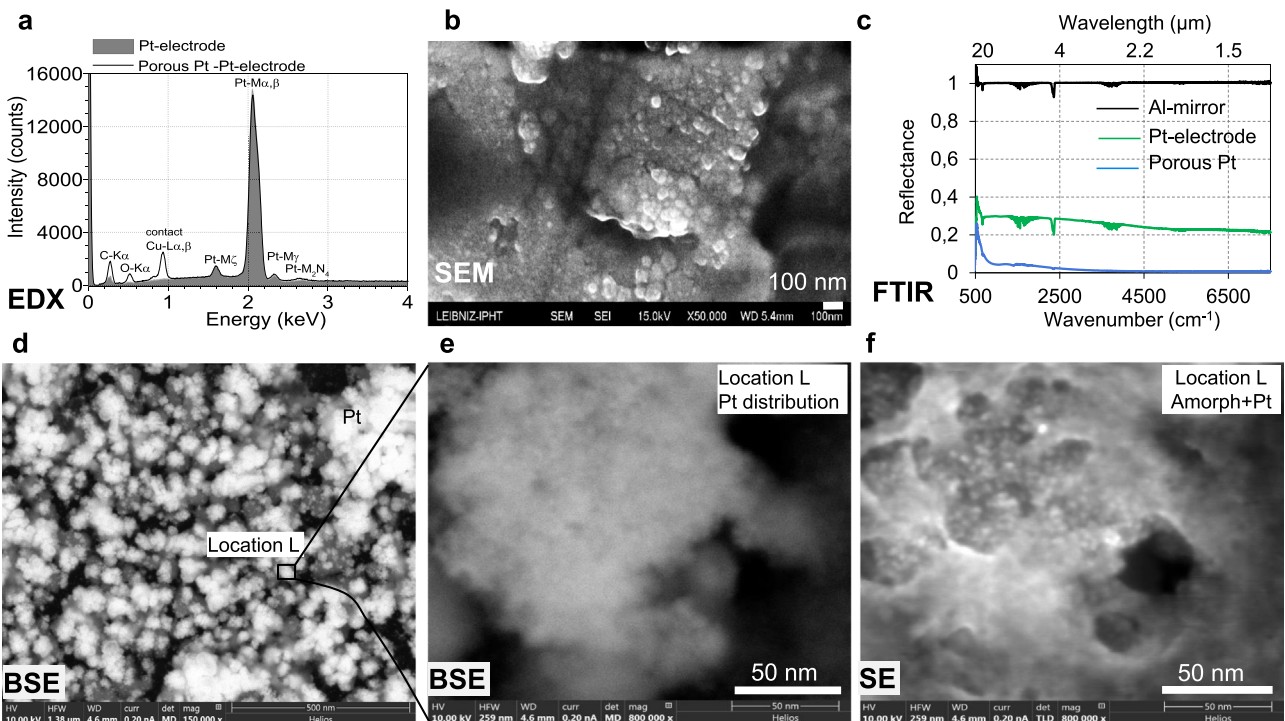

**Fig. 3 EDX and BSE reveals platinum crystals inside the layer. a** EDX diagram of porous platinum film on platinum cathode recorded on the area indicated in (**b**); **b** SEM image of porous platinum film used for EDX analysis; **c** FTIR reflectance spectra in the wavenumber region 7500–500 cm⁻¹ of the platinum porous at 250 s electrolysis (blue lines) compared with the bar platinum electrode (green) and Al mirror (black line). **d–f** BSE (**d**, **e**) and SE (**f**) SEMs of the platinum layer; **e**, **f** images are recorded on the same location indicated in (**d**); BSE mode shows a dark grey level for the light elements such as silicon, carbon and bright spots for heavier elements such as platinum.

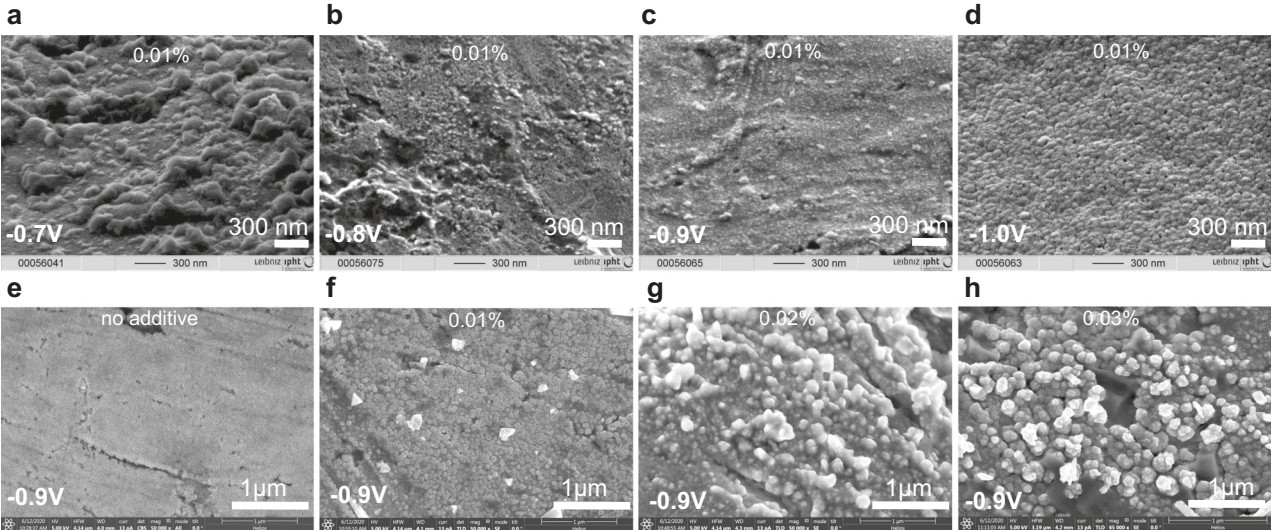

**Fig. 4 Applied potential determines the porosity of platinum. a–d** SEM images of the porous platinum obtained from 0.5% PtCl$_4$, 0.01% Pb(CH$_3$COO)$_2$ in isopropanol at different applied potential; **e–h** SEM images of the porous platinum obtained at −0.9 V from 0.5% PtCl$_4$ in isopropanol at different concentration of additive: 0% (**e**), 0.01% (**f**), 0.02% (**g**) and 0.03% (**h**) Pb(CH$_3$COO)$_2$.

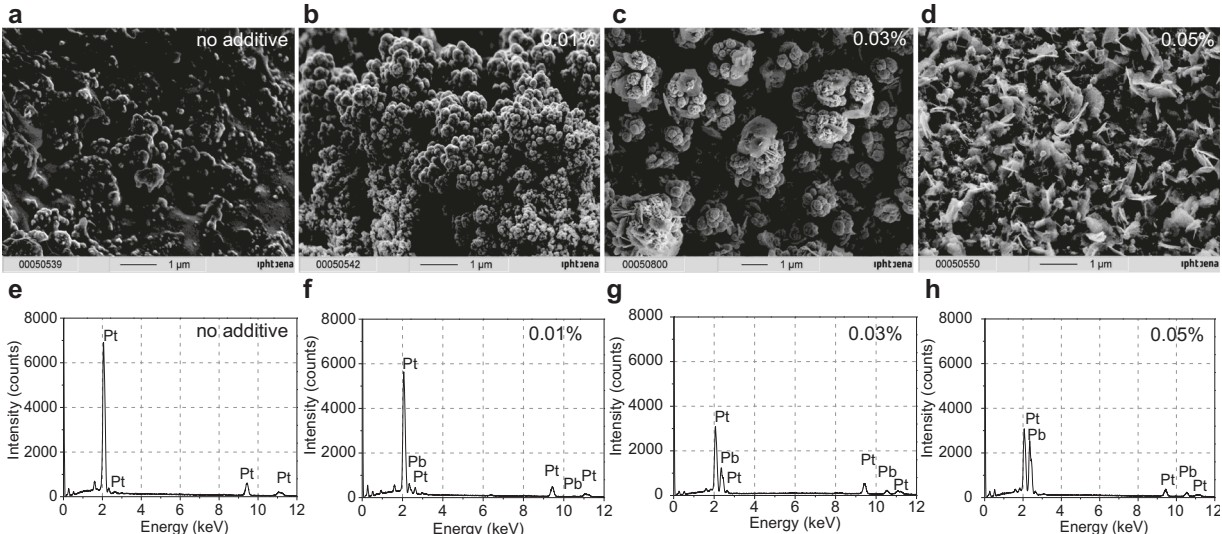

**Fig. 5 Porous platinum layer through aqueous electrochemistry. a–d** SEM image of the porous platinum obtained at −1 V from aqueous 0.5% PtCl$_4$ at 0% (**a**), 0.01% (**b**), 0.03% (**c**), 0.05% (**d**) Pb(CH$_3$COO)$_2$; **e–h** EDX of the porous layer at 0% (**e**), 0.01% (**f**), 0.03% (**g**), 0.05% (**h**) Pb(CH$_3$COO)$_2$. Lead is removed from the porous platinum layer by dissolution with HClO$_4$ or by sublimation at 600 °C.

rounded and some of them are tri- to polyangular. TEM images recorded on the mature porous platinum layer show several types of particles at the margins of the layer, some of them are amorphous and some are crystalline in the core with an amorphous shell. The form of the amorphous particles tends to be round, crystalline ones tend to be elongated. These shapes are similar to the ones observed in SEM images (Fig. 1b, c), in which the amorphous parts alternate with crystalline ones (Fig. 3d–f). In addition, the FFT of the matured porous layer in Fig. 2 shows that crystalline particles are not single crystals, they contain defects (i.e. twins).

**Layer porosity correlated to hydrogen evolution at the electrode.** Changes in porosity of the platinum layer as a consequence of changes in mass transfer and electrical potential profile across the electrode electric double layer are expected. The electric

potential profile-shaped by the charge of the inner Helmholtz layer (IHL) ($\sigma^{IHL}$), the charge of the outer Helmholtz layer (OHL) ($\sigma^{OHL}$) and the charge of diffuse layer ($\sigma^d$) (Fig. 10) determines the platinum ions reduction event and the co-generation of hydrogen evolution during the metal formation. To create a nanoporous electrode, the co-generation of hydrogen along with metal deposition, represented a scientific interest for a long time[1–3]. This co-generation of hydrogen during the platinum ion reduction takes place at the electrochemical generation of platinum black at the cathode in aqueous media[1–3] or in non-aqueous media[7]. The role of hydrogen abundance at the cathode during the metal deposition is directly connected with the porosity of the metal foam formation. There is a recent review that compiles the factors of influence (electrolyte composition, temperature and applied potential) at the formation of foam using the hydrogen bubble templating method[16]. Aligned to this idea, that the

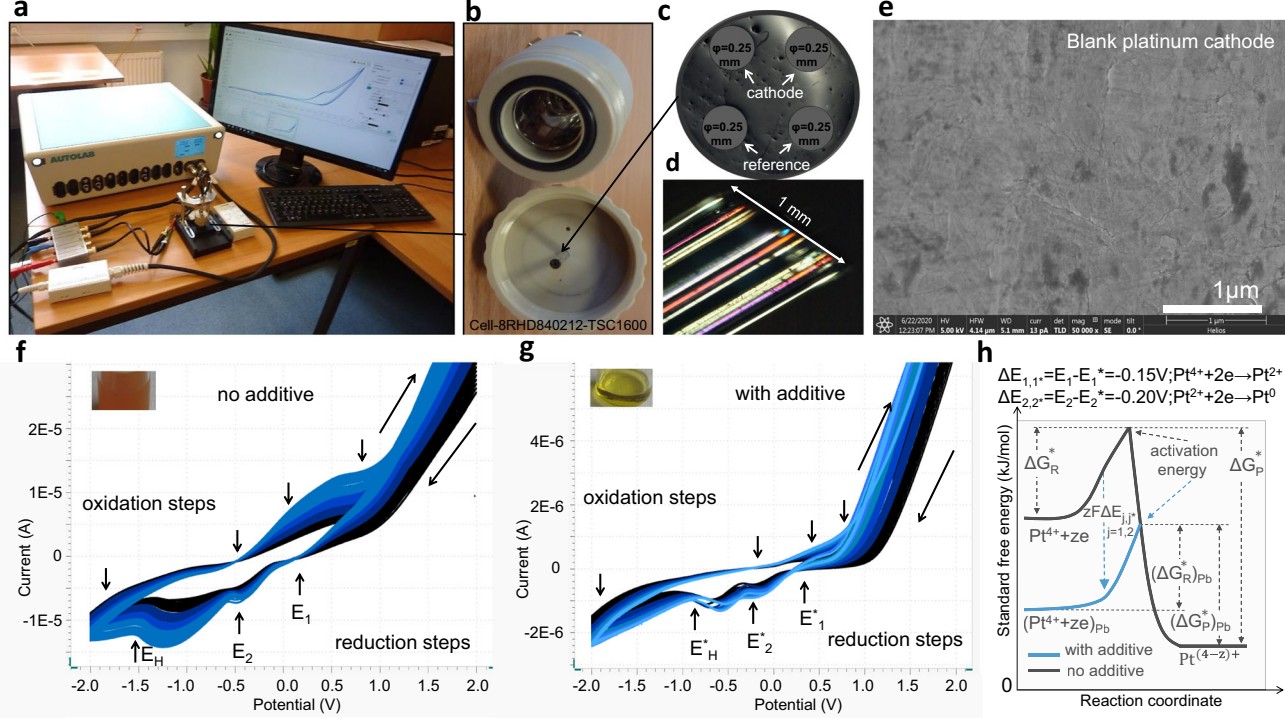

**Fig. 6 Cyclic voltammetry study of PtCl₄ in isopropanol. a–c** Setup (**a**) 8RHD840212 cell (**b**) equipped platinum-electrodes (**c**); **d** light microscopy lateral view of the glass embedded platinum-electrodes; **e** SEM image of the blank platinum electrode; **f**, **g** CVs of 0.5% $PtCl_4$ in isopropanol recorded at 0.1 V/s in the absence of 0.01% $Pb(CH_3COO)_2$ (**f**) and in the presence of 0.01% $Pb(CH_3COO)_2$ (additive) (**g**). CV starts at +2V (**f**, **g**). For $Pt^{4+} + 2e^- \rightarrow Pt^{2+}$, $E_1 = +0.10$ V (without additives) and $E_1^* = +0.25$ V (with additives), for $Pt^{2+} + 2e^- \rightarrow Pt^0$; $E_2 = -0.4$ V (without additives) and $E_2^* = -0.2$ V (with additives); **h** diagram of the standard free energy during the platinum ion reduction activation. Standard free energy for the reactant $Pt^{4+}$, $\Delta G_R^*$, $(\Delta G_R^*)_{Pb}$ and for the product $Pt^{(4-z)+}$, $\Delta G_P^*$, $(\Delta G_P^*)_{Pb}$ in the absence and presence of lead, and the energy amount, $zF\Delta E_{j,j^*}$, $(j = 1, 2)$ with which the reaction curve is lowered by lead are indicated on the panel.

abundance of hydrogen increases at the more negative potential (i.e. from −0.7 to −1 V), we observed an accentuated dependence of the porosity on the applied potential with all other electrolysis conditions being identical: at −0.8 V, the pore size was 20 nm; at −0.9 V, it became 35 nm; at −1 V, it was 50 nm (Fig. 5, Fig. 10c–e, Supplementary Table 1). The pore sizes of the sponge platinum in aqueous electrochemistry are 100 times larger and can reach 500 nm (Fig. 6). One can consider two levels of porosities: (1) the pores inside the porous platinum structure (Fig. 3) and (2) the pores between the porous platinum structures (Figs. 4 and 5). As an example, in this work, we demonstrated how the structure of the nanoporous platinum layers can be controlled by the potential and by the electrolyte composition (aqueous and non-aqueous media). For a given potential and electrolyte composition the thickness is well controlled by the electrodeposition time. In Supplementary Table 1, we list results from the thickness control using potential, electrolyte composition and electrodeposition time. To analyse the layer thickness and porosity we used Focus Ion Beam cut and SEM imaging of the layer transversal cut or of the layer scratches. Supplementary Table 1 shows that the porosity of the platinum layer increases in both aqueous and non-aqueous media at the more negative potentials from −0.7 to −1.1 V. A plausible explanation to this phenomenon can be that the more abundant hydrogen evolution at more negative potential increases the entropy of the system and the structures are less compact. The absence of hydrogen and additives during the electroreduction of platinum ion causes the formation of face centred cube (fcc) metallic platinum, however, their presence origins a disruption in the metallic layer formation, although the metallic crystals remain in the structure at a different orientation,

sizes and positions creating a disorder and porosity. From XRD analysis, the lattice of the fcc is 3.9231 Å, which fits the database value of the metallic platinum fcc lattice. We assume that the nanoscale crystallite contains platinum fcc unit cells (Fig. 1a).

EDX and XRD of the porous platinum formed on platinum electrodes confirm the degrees of purity and crystallinity (Figs. 1d and 3a). EDX does not show the presence of lead in the porous platinum layer obtained in non-aqueous media, only a weak Pb-La emission at 10.6 keV. However, the presence of lead with the lines Pb-La,b in platinum porous layer obtained in aqueous media is indicated by EDX (Fig. 5f–h) and SEM (Fig. 5c, d). The EDX peak of Cu-Lα,β impurity derives from electrode contacts (Fig. 3a, Fig. 5e–h). Lead is removed from the porous platinum layer by dissolution with HClO₄ or by sublimation at 600 °C[3]. Fourier Transform Infra-red spectra have been recorded for the cathode (Fig. 3c), before (green line) and after platinum deposition (blue line) show low reflectance of the porous platinum layer in the wavenumber region 500 cm⁻¹ ÷ 7500 cm⁻¹, in agreement with our previous reported data[7].

## Discussion

**Qualitative insight into the mechanism of the porous platinum layer electrochemical growth.** The deposition starts in certain regions on the surface, where a cauliflower-like structure with a height of approximately 20 nm appears, which subsequently extends to cover the entire surface[7]. The initiation of platinum black can occur at the cathode defects; however, one cannot exclude that this effect results from unusual electric field interferences. This initiation can also be affected by the adsorption competition between platinum ions and electrolyte molecules. To

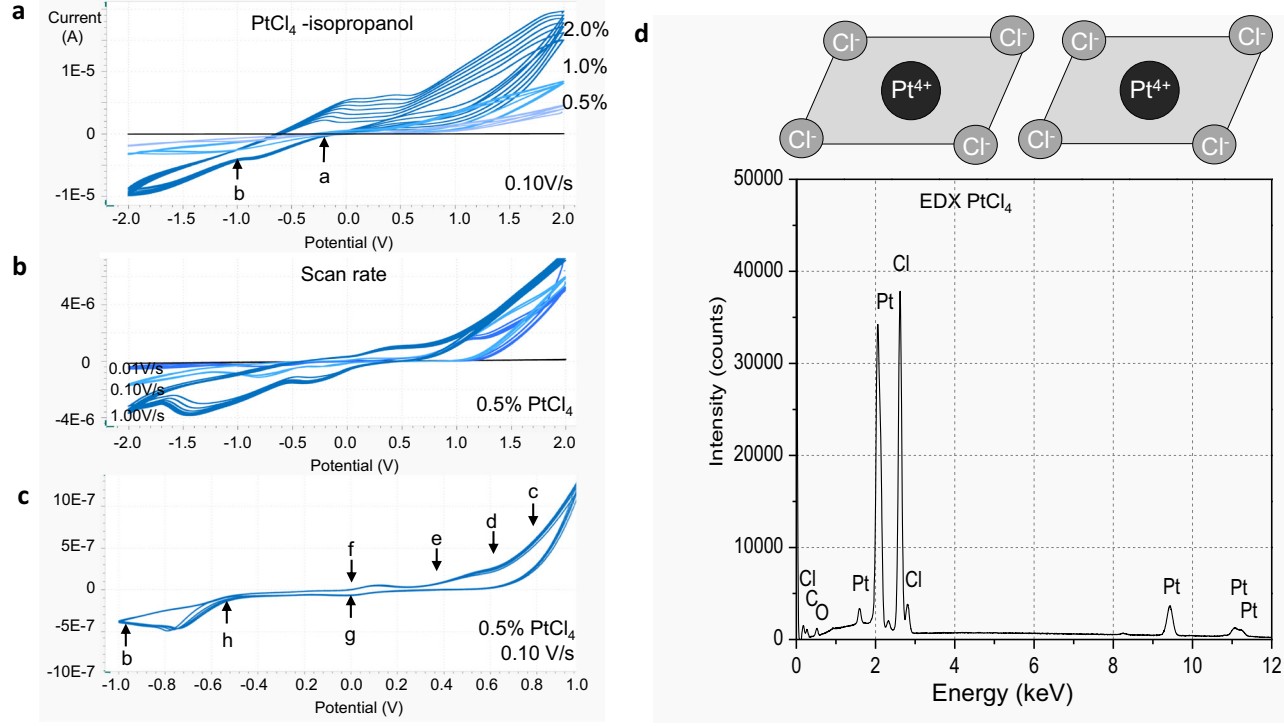

**Fig. 7 Evolution CVs of PtCl$_4$ in isopropanol. a–c** CVs of PtCl$_4$ in isopropanol: **a** 0.5% PtCl$_4$, 1% PtCl$_4$, 2% PtCl$_4$, 0.1 V/s, start potential +2 V and end potential −2 V vs. platinum reference electrode; **b** 0.5% PtCl$_4$ at 0.01 V/s (blue line), 0.1 V/s (light blue line) and 1 V/s (dark blue line); **c** CVs of 0.5% PtCl$_4$ recorded at 0.1 V/s, start potential +2 V and end potential −2 V vs. platinum reference electrode. The reactions: **a** Pt$_{4+}$ + 4e → Pt; **b** 2H$^+$ + 2e → H$_2$; **c** 2(CH$_3$)$_2$C–OH→2(CH$_3$)$_2$C=O + 2H$^+$ + 2e$^-$; **d** 2Cl$^-$ → Cl$_2$ + 2e; **e** Pt$^{2+}$ → Pt$^{4+}$ + 2e; **f** Pt → Pt$^{2+}$ + 2e; **g** Pt$^{4+}$ + 2e → Pt$^{2+}$; **h** Pt$^{2+}$ + 2e → Pt; **d** The EDX of the PtCl$_4$, chemical used for electrosynthesis of porous platinum.

find whether the electrochemical reaction implies a direct four-electron platinum ion reduction or a multi-step process and how the cathodic potential determines the initiation and formation of the flat and porous platinum nanolayers, we used transient and reverse electrochemical methods.

**Non-aqueous medium CVs.** By analysing the cyclic voltammograms of 0.5% PtCl$_4$ in isopropanol (Fig. 6f), one observes three well-defined waves of reduction at a large sweep rate of 1 V/s and two waves of oxidation that were better observable at low sweep rates of 0.01 and 0.1 V/s (Fig. 7b). An irreversible reduction wave from −0.5 to −1 V (at 0.01 V/s) to −1.1 V (at 0.1 V/s) to −1.8 V (at 1 V/s) is clearly observed on the CVs. The CV displays a redox tail of chlorine gas (2 Cl$^-$ → Cl$_2$ + 2e$^-$) and acetone formation [2(CH$_3$)$_2$C–OH → 2(CH$_3$)$_2$C=O + 2H$^+$ + 2e$^-$] in the interval 1 ÷ 2 V at a sweep rate of 1 V/s, and at lower potentials of 0.8 V, 0.9 V at slower sweep rates (0.1 and 0.01 V/s) when their separation is visible (Fig. 7b). The system in Figs. 6f and 7a–c does not contain additives and does not generate adherent porous platinum. In contrast, the investigated system the Fig. 6g contains additive and generates adherent platinum black. We observed that the additive considerably changes the CV shape, even if the additive alone does not show prominent electroactivity at the used concentration (Fig. 8a, grey line). A small reduction wave starting at −0.2 V and an oxidation wave starting at 0.6 V can be identified on the I–E curve. We recorded a shift of the platinum ion oxido-reduction peaks at a more positive potential (Fig. 6g) and an increase in current intensities due to the additive kinetic involvement. The lead additive is not observed in the XRD diagram of platinum deposits, which implies that platinum does not concurrently grow with lead. The additive can also inhibit hydrogen formation and increase the coulometric efficiency of platinum layer growth[17]. In Fig. 6g, four reduction waves are

identified on the CVs, which were shifted towards the positive potential compared to those in Fig. 6f. The electrode surface might change because the deposition causes deviations in the platinum ion reduction peak (the peaks of the black, dark blue, blue and light blue waves appear at −0.7, −0.65, −0.6 and −0.55 V, respectively, in Fig. 6g). The following hypothesis is true but not clearly sustainable: at the cathode, the direct four-electron reduction of platinum ions (Pt$^{4+}$ + 4e$^-$ → Pt↓) with gaseous hydrogen formation is more probable for the first system (Fig. 6f) than for the second one (Fig. 6g). A metastable four-electron transfer of platinum oxido-reduction centred at −0.5 V was identified in Fig. 7a; however, with the addition of PtCl$_4$, the wave of four-electron oxidation splits into two waves of two-electron oxidation. The CVs in Fig. 6g supports a two-step mechanism: Pt$^{4+}$ + 2e$^-$ → Pt$^{2+}$ and Pt$^{2+}$ + 2e$^-$ → Pt↓. In Figs. 6f and 7b, the evolution of the hydrogen reduction peak is visible between −0.75 and −1.5 V, but this peak is absent in Fig. 6g. Here, the peak is replaced by a tail of reduction in this interval, which can be attributed to the hydrogen ion reduction. The wave of oxidation in the interval of 0.8 ÷ −0.5 V assigned to the platinum oxidation, is no longer visible in the system containing additive (Fig. 6g), which suggests that platinum ions have undertaken an irreversible reduction or that the oxidation of platinum ion species shifts to higher potential values. To summarise the observations on the cyclic voltammograms at the reduction of platinum ions in a non-aqueous medium (Figs. 6 and 7), one can affirm that the peaks in reduction steps without additives are at more negative potential than peaks in reduction steps with additives. For the first step of reduction Pt$^4$ + 2e$^-$ → Pt$^{2+}$ the electrochemical potentials are $E_1$ = +0.10 V (without additives) and $E_1^*$ = +0.25 V (with additives) resulting in a potential difference $\Delta E_{1,1^*} = E_1 - E_1^* = -0.15$ V. For the second step of reduction Pt$^{2+}$ + 2e$-$ → Pt$^0$; $E_2$ = −0.4 V (without additives)

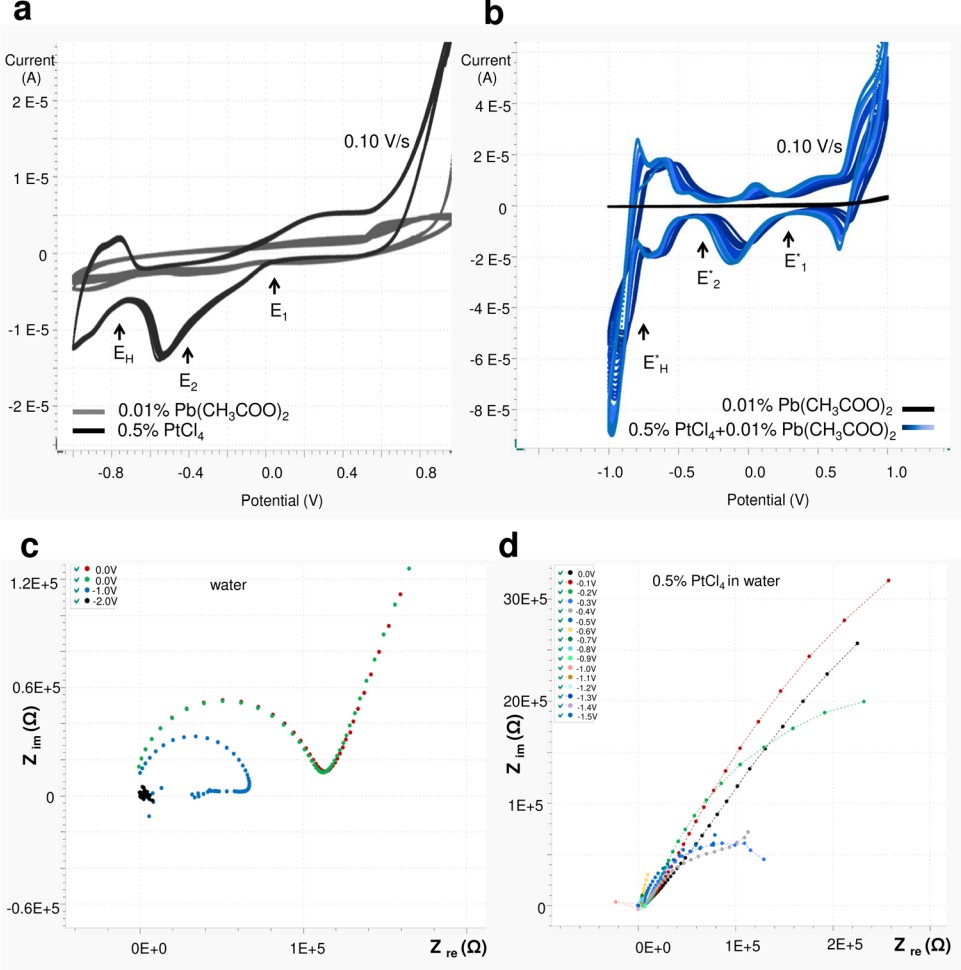

**Fig. 8 Electrochemistry of PtCl$_4$ CVs in water. a** Separately recorded CVs of aqueous 0.5% PtCl$_4$ (black line) and 0.01% Pb(CH$_3$COO)$_2$ (grey line); **b** separately recorded CVs of aqueous 0.01% Pb(CH$_3$COO)$_2$ and 0.5% PtCl$_4$ + 0.01% Pb(CH$_3$COO)$_2$ (nuanced blue lines). For Pt$^{4+}$ + 2e− → Pt$^{2+}$, $E_1$ = +0.05 V (without additives) and $E^*_1$ = +0.35 V (with additives), $\Delta E_{1,1*}$ = $E_1$ − $E^*_1$ = −0.30 V. For Pt$^{2+}$ + 2e− → Pt$^0$; $E_2$ = −0.4 V (without additives), $E^*_2$ = −0.25 V (with additives), $\Delta E_{2,2*}$ = $E_2$ − $E^*_2$ = −0.15 V; **c, d** Nyquist plots of water at 0 V (red, green points), −1 V (blue points) and −2 V (black points) (**c**) and 0.5% PtCl$_4$ in water at 0, −0.1, −0,2, −0.3, −0.4, −0.5, −0.6, −0.7, −0.8, −0.9, −1, −1.1, −1.4, −1.2, −1.3, −1.4 and −1.5 V (**d**).

and $E^*_2$ = −0.2 V (with additives), the potential difference is $\Delta E_{2,2*}$ = $E_2$ − $E^*_2$ = −0.2 V. This lowers the reaction curves, respectively the free activations energies with $−zF\Delta E_{1,1*}$ = 28.9 kJ/mol for the first step and with $−zF\Delta E_{2,2*}$ = 38.6 kJ/mol for the second step.

**Aqueous medium CVs.** The aqueous electrochemistry is governed by higher values of current intensities and well-defined redox peak. Mostly regulated by the electrical potential gradients and ions mobility, the current density shows higher values compared to the non-aqueous system, at a similar concentration of PtCl$_4$. This means a quicker mass transfer of Pt$^{4+}$ species to the cathode deduced from quicker migration, diffusion and convection of ions from solution to the electrode. The currents recorded in the presence of additive have higher values, which interpret the electrical potential gradients increase as a consequence of thermodynamic parameters (i.e. enthalpy) gradient increase. The staircase CV of PtCl$_4$ 0.5% in water is plotted in Fig. 8a (black line) and shows a prominent peak of platinum reduction, one peak of hydrogen ions reduction at −1 V and two oxidation shoulders: the oxygen ion oxidation partially overlapped with the chloride oxidation (+1 V). The presence of additive (Fig. 8b) generates a completely changed aqueous CV with four well-defined peaks of oxidation and four peaks of

reductions, which show different kinetics and a clear separation between hydrogen reduction and platinum ion reduction. Using the procedure from Feltham et al.[3], we estimated the rate constant of the first-order reactions: Pt(IV) + 2e → Pt(II) at a standard rate constant of $1.4 \times 10^{-5}$ cm s$^{-1}$ and Pt(II) + 2e → Pt (0) at the standard rate constant of $2.2 \times 10^{-5}$ cm s$^{-1}$. The corresponding reduction potentials are more negative than those reported by Feltham et al.[3].

For the aqueous electrochemistry, the CVs data show a clear electro-kinetic difference between PtCl$_4$/water (no platinum porous) and PtCl$_4$/water/additive (platinum porous) (Fig. 8). The additive shifts the redox potential of platinum ions to more positive potentials and increases the platinum ion reduction rate. For the first step of reduction Pt$^{4+}$ + 2e$^−$ → Pt$^{2+}$ the electrochemical potentials are $E_1$ = +0.05 V (without additives) and $E^*_1$ = +0.35 V (with additives) resulting in a potential difference $\Delta E_{1,1*}$ = $E_1$ − $E^*_1$ = −0.30 V. For the second step of reduction Pt$^{2+}$ + 2e$^−$ → Pt$^0$; $E_2$ = −0.4 V (without additives) and $E^*_2$ = −0.25 V (with additives), the potential difference is $\Delta E_{2,2*}$ = $E_2$ − $E^*_2$ = −0.15 V. This lowers the reaction curves, respectively the free activations energies with $−zF\Delta E_{1,1*}$ = 57.9 kJ/mol for the first step and with $−zF\Delta E_{2,2*}$ = 28.9 kJ/mol for the second step.

For both aqueous and nonaqueous medium, the addition of lead ions causes a shift of the platinum redox potential. An

explanation of this shift can be related to the larger ionic radius of the $Pb^{2+}$ $(r = 119\,pm)$[18] additives in comparison to the ionic radius of $Pt^{4+}(r = 62.5\,pm)$[18], which causes an imbalance of the charges at the interphase. Furthermore, the large ionic radius of $Pb^{2+}$ makes local changes related to complexation, ion-pairing and ionic strength variation[8].

To correlate the electrode potential shift with the thermodynamic properties of the system in the presence of lead, we start from the definition of electrochemical potential. The electrochemical potential at the interface of the two phases makes to take place the reaction of platinum reduction indicated in Eq. (1)

$$Pt^{4+} + ze^- \rightarrow Pt^{(4-z)+} \qquad (1)$$

and the electrode process involves fast charge transfer governed by the Nernst equation.

By definition, the electrochemical potential, for species platinum with the charge 4+ and inner potential $\phi$ in phase metal (M), $\overline{\mu_{Pt}^M}$, respectively in the phase solution (S), $\overline{\mu_{Pt}^S}$, is defined by the Eq. (2)

$$\overline{\mu_{Pt}^M} = \mu_{Pt}^M + zF\varphi^M \text{ and } \overline{\mu_{Pt}^S} = \mu_{Pt}^S + zF\varphi^S \qquad (2)$$

In the Eqs. (3) and (4), the chemical potential for the same species in phase M, $\mu_{Pt}^M$, respectively, in phase S, $\mu_{Pt}^S$, has the correspondence to thermodynamic parameters

$$\mu_{Pt}^M = \left(\frac{\delta G}{\delta n_{Pt}}\right)_{T,P} = \left(\frac{\delta H - T\delta S}{\delta Pt}\right)_{T,P} \qquad (3)$$

$$\mu_{Pt}^S = \left(\frac{\delta G}{\delta n_{Pt}}\right)_{T,P} = \left(\frac{\delta H - T\delta S}{\delta Pt}\right)_{T,P} \qquad (4)$$

where $n_{Pt}$ is the number of moles of $Pt^{4+}$ in the phase M or S; G, H and S are the thermodynamic parameters: free energy, enthalpy and entropy, respectively.

At metal/solution interphase equilibrium (eq), the equality between the electrochemical potentials of the species $Pt^{4+}$ in the two phases is achieved as indicated in Eq. (5)

$$[\bar{\mu}_{Pt}^M = \bar{\mu}_{Pt}^S]_{eq} \qquad (5)$$

By combining and rearranging the Eqs. (3)–(5), the relationship described in the Eqs. (6) and (7) can be written

$$\left[\left(\frac{\partial H - T\partial S}{\delta n_{Pt}}\right)^M_{T,P} + zF\varphi^M\right]_{eq} = \left[\left(\frac{\partial H - T\partial S}{\delta n_{Pt}}\right)^S_{T,P} + zF\varphi^S\right]_{eq} \qquad (6)$$

$$\left[\left(\frac{\partial H - T\partial S}{\delta n_{Pt}}\right)^S_{T,P} - \left(\frac{\partial H - T\partial S}{\delta n_{Pt}}\right)^M_{T,P}\right]_{eq} = \left[zF(\varphi^M - \varphi^S)\right]_{eq} = zFE_{eq}$$

chemical term                          electrical term

$$(7)$$

where the potential difference between the metal/solution phases expresses the electrode potential (E). Moreover, $zF(\varphi^M - \varphi^S)$ in Eq. (7) represents the electrical component of the free energy (G), respectively of the work/energy necessary to transfer z electrons across the metal/solution interface[19]. The left part of Eq. (7) shows how the thermodynamic parameter change (H, S) influence the redox potential. In this context, Pb modifies the interphase thermodynamic equilibrium and charge equilibrium. On one hand, the Pb ions cause changes in the interfacial potential difference by altering the charge balance and the charge density at the interface, due to a larger ionic radius (119 pm for $Pb^{2+}$ compared to 62.5 pm for $Pt^{4+}$)[18]. On the other hand, the electrochemical potential of each phase depends on the associated enthalpy and entropy. Therefore, the interfacial potential

differences can also occur without charge excess at the interface as a result of the equation of definition (Eqs. 2–7). The interfacial potential difference (Eq. 7) determines the electro-kinetic (fast charge transfer) event at the electrode, being complexly influenced by the thermodynamic parameters at the boundary metal-solution. Figure 6h shows the effect of potential change on the standard free energies during the platinum reduction indicated in Eq. (1), as a consequence of lead presence. The lead moves the electrochemical potential to more positive values with ΔE in both aqueous and non-aqueous medium. The relative energy of the electrons on the cathode also changes with $-zF\Delta E$ and the reaction curve moves down with this amount of energy, with which the activation energy of the reaction is lowered. It is realistic to accept that lead ions modify the interfacial entropy and enthalpy (increase in enthalpy due to the binding forces increase) with consequences upon the interfacial electrical equilibrium.

**Non-aqueous EIS**. The total impedance (Z) between the electrodes[12–15] was measured to reveal complementary details on the platinum formation mechanisms. The complex impedance Z is defined in the Eq. (8) as the ratio of the complex voltage ($\bar{V}$) and current ($\bar{I}$) intensities of wave phase $\Phi$ difference from the input[8,9]:

$$Z = \frac{\bar{V}}{\bar{I}} = \frac{V_0}{I_0}e^{-i\Phi} \qquad (8)$$

The regions of mass transfer (at low frequencies) and kinetic (at high frequencies) are sketched in Fig. 10a, at the coordinate of the imaginary impedance ($Z_{im}$ related to the capacitance) and real impedance ($Z_{re}$ related to the resistance). Z is fragmented into the following components: charge transfer resistance $R_{ct}$, which expresses the kinetics of the heterogeneous charge transfer, and the components of Warburg impedance, $R_w$ and $C_w$, which exhibits diffusional mass transfer. In Fig. 10a, the semicircular region of the Nyquist plot shows the coupling between double-layer capacitance and electrode kinetic effects at higher frequencies than the diffusion process. The diagonal diffusive tail relates to the diffusion at low frequencies[13–15]. At 0 V, we observe a capacitive tail with Warburg behaviour of 45°; at −0.5 V, the angle became smaller than 45° (Fig. 10). Hence, the chemical reactions couple with a slower electron transfer and are progressive from −0.1 to −0.5 V. The effect of slowing the electron transfer is accentuated in the absence of additive. This insight supports the role of the additive in the platinum ion reduction rate. Moreover, at higher voltages (Fig. 10a), the impedance plot shapes in a semicircle connected to make an almost complete circle. According to the literature[13], one attributes the curve curling to the weak adherence and detachment of the layer. In this heterogeneous system, if one phase has a low volume fraction, and low conductivity, two separate shapes of the impedance can be found in the response (i.e., one large-diameter semicircle coupled to a small-diameter semicircle). If the phase at low concentration has higher conductivity than the other one, two semicircles will apear[13–15], as observed in Fig. 9d at a more negative potential than −0.6 V. Instead, the presence of the additive (Fig. 9e) increases the resistance and capacitance by approximately one order of magnitude along with a considerable shape change. The impedance representation at low frequencies shows an increase in phase angle (almost 90°) (Figs. 9b and 10b) at the potentials from −0.8 to −1.1 V (Fig. 9b), where the dissolved species of platinum ions are electroactive. This indicates the electrode layer involvement in the reversible charge transfer. In the potential interval from −0.8 to −1.1 V, we assist a phase angle change from

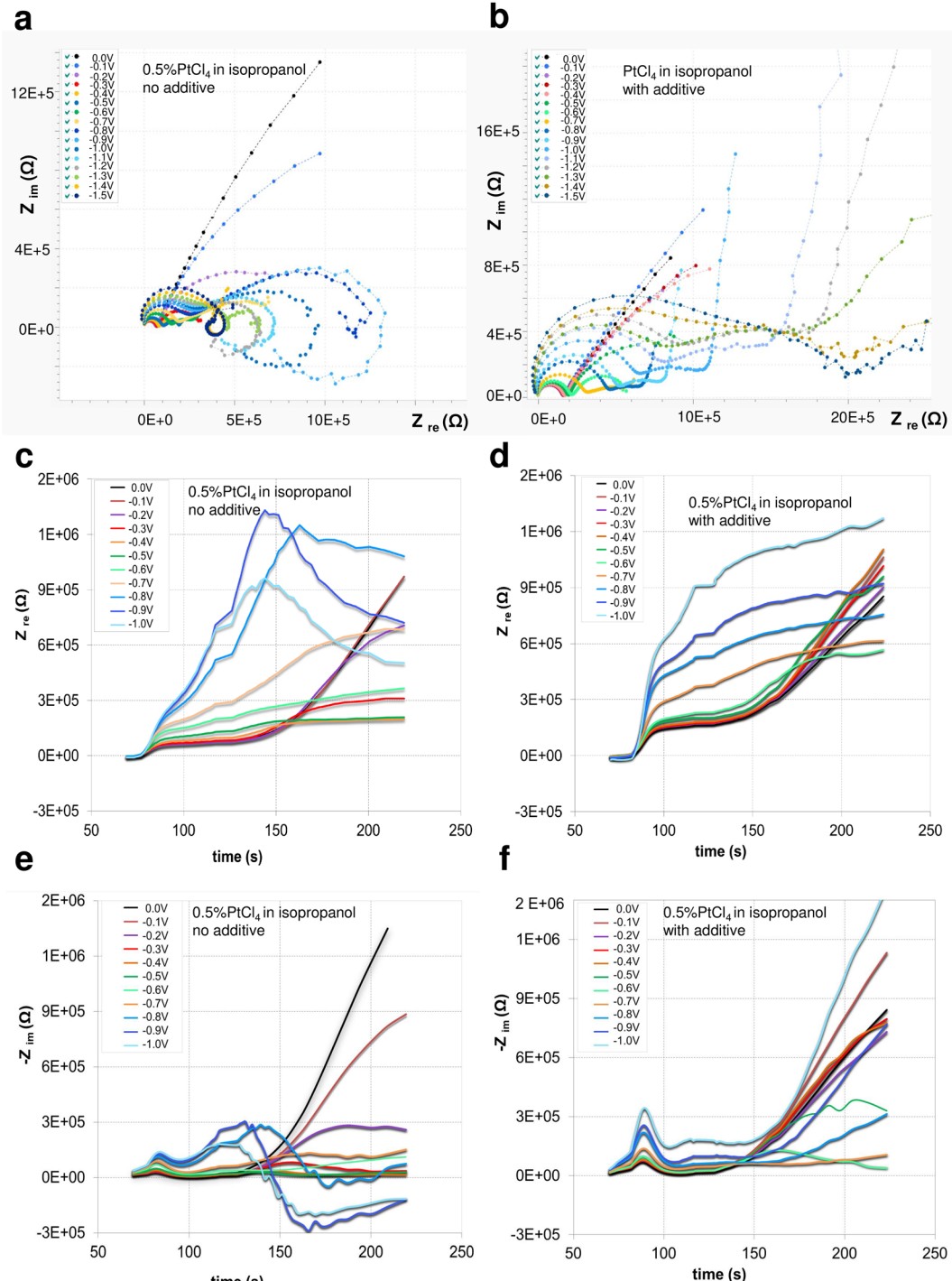

**Fig. 9 Nyquist plots. a** 0.5% PtCl$_4$ isopropanol; **b** 0.5% PtCl$_4$ isopropanol 0.01% Pb(CH$_3$COO)$_2$; 0 to −2 V vs. reference, VRMS = 10 mV, frequency range 0.1 ÷ 10$^5$ Hz; **c–f** impedance-time data representation 0.5% PtCl$_4$ isopropanol (**c**, **e**) and 0.5% PtCl$_4$, 0.01% Pb(CH$_3$COO)$_2$ in isopropanol (**d**, **f**). 0 to −1 V vs. reference, VRMS = 10 mV, frequency range 0.1 ÷ 10$^5$ Hz.

almost 0° to almost 90°, i.e., after the chemical reaction coupled with a very slow electron transfer (0°), we switch to a reversible charge transfer between the electroactive species and the formed layer at the electrode (90°).

**Aqueous EIS**. At similar parameters used for isopropanol, the Nyquist plots were recorded in water (Fig. 8c, d). Qualitatively, they show that the electroreactions in water are approximately one order of magnitude faster than in isopropanol. At frequencies

larger than 1000 Hz, the kinetic involved is even faster than the frequency change. This determines the shape of the Nyquist plots, which does not display closed semicircles. We demonstrate in Fig. 10b–e that the porous platinum layer obtained from non-aqueous electrodeposition exhibit characteristics of pore size and resistance, which increase with the applied potential from 25 nm ($Z_{re}$ = 0.7 MΩ) at −0.8 V to 35 nm ($Z_{re}$ = 0.8 MΩ) at −0.9 V and to 80 nm ($Z_{re}$ = 1.6 MΩ) at −1.1 V. The electrochemical deposition of porous platinum nanolayers was limited and conducted at

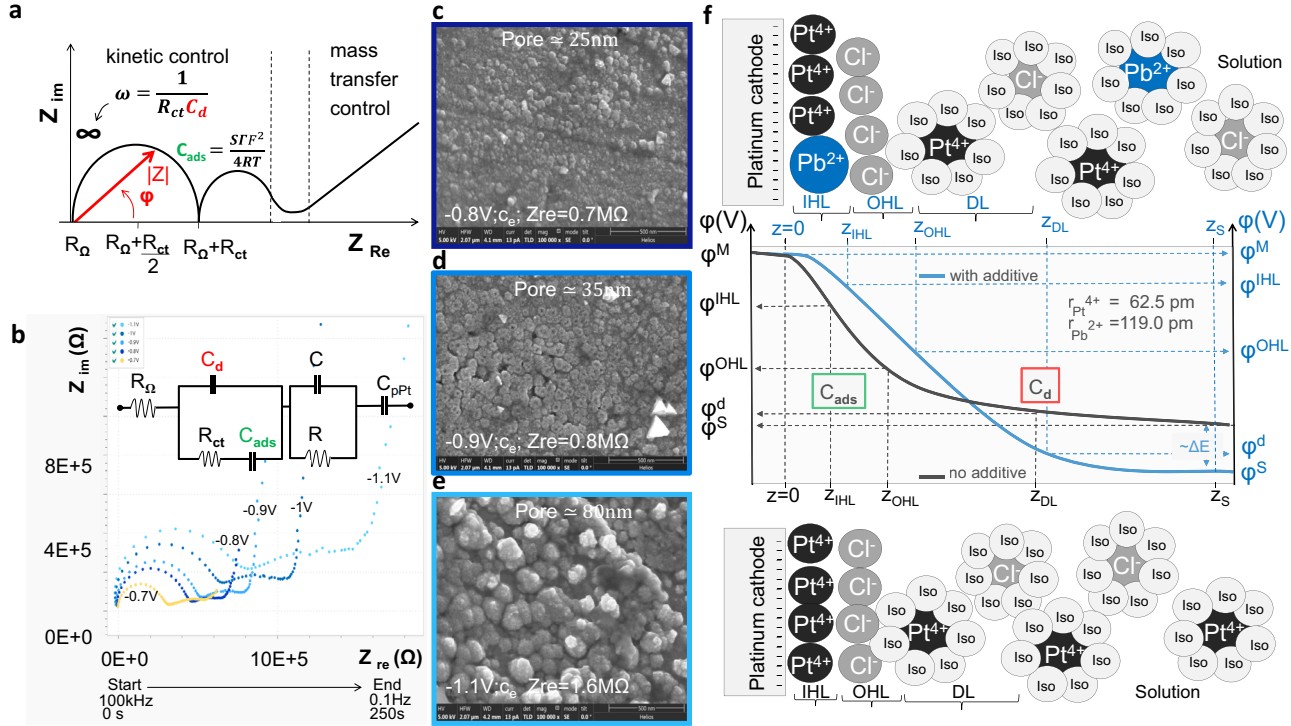

**Fig. 10 Porous platinum formation viewed in Nyquist plots. a** Sketch of total impedance ($|Z|$), where $R_\Omega$ = solution resistance, $C_d$ = double-layer capacitance, $R_{ct}$ = charge-transfer resistance, $\omega = \frac{1}{R_{ct}C_d}$ and $C_{ads}$ = capacitance of the adsorbed layer; $C_{ads} = \frac{S\Gamma F^2}{4RT}$, $\Gamma$—the adsorbed layer amount (mol/cm²); F—Faraday constant, R—gas constant, S—electrode area, T—absolute temperature; **b** overlay of Nyquist plots of Ce = 0.5% $PtCl_4$ isopropanol 0.01% $Pb(CH_3COO)_2$ at −0.7, −0.8, −0.9, −1.0 and −1.1 V; **c–e** SEMs of the porous platinum layer obtained from 0.5% $PtCl_4$ isopropanol 0.01% $Pb(CH_3COO)_2$ −0.8 V (**c**), −0.9 V (**d**) and −1.1 V (**e**); **f** Diagram of electric double layer formation at the electrode correlated with the electrochemical potential difference ($\Delta E < 0$) generated by the additive presence. The electric double layer is formed of the inner Helmholtz layer (IHL), outer Helmholtz layer (OHL) and diffuse layer (DL). The corresponding electrical potential profile across the electrode electric double layer: $\varphi^M$-inner potential of species in the metal phase (V) corresponding to $z_M$, at the distance z = 0 from electrode, $\varphi^S$-inner potential of species in the solution phase (V) corresponding to $z_S$; $\varphi^{IHL}$ potential of inner Helmholtz layer (V) corresponding to $z_{IHL}$; $\varphi^{OHL}$ potential of outer Helmholtz layer (V) corresponding to $z_{OHL}$, $\varphi^{DL}$ potential of diffuse layer (V) corresponding to $z_{DL}$; blue colour marks the system in the presence of the additive. The corresponding positions to the $C_{ads}$ and $C_d$ are indicated.

room temperature (22 °C). The temperature increase can modify the electrochemical behaviour of the involved chemical species and electrode stability causing changes in layer thickness and morphology[20].

**Application of porous platinum nanolayers**. Electrosynthesized porous platinum has characteristics of a perfect broadband absorber material and confers it application as a high-temperature resistant optical absorber material in thermo-sensors. By adjusting thicknesses and porosity of the layer the optical absorbance can be fine-tuned and make electrosynthesized porous platinum also interesting for plasmonics. The large surface-area-to-volume ratio of highly porous platinum nanolayers determine their catalytic and electrocatalytic properties and are of advantage for implementation in fuel-cell, chemical, petrochemical and pharmaceutical industry. Furthermore, electrodes coated with porous platinum nanolayers can be used in high-resolution electrochemical sensing due to the efficient electric current exchange and excellent structural stability of porous platinum[21]. Due to the large chemical and thermal stability of porous platinum nanolayers with well-controlled thickness and porosity this material is advantageous for use as shelter-protection of metals with high chemical reactivity in pyro-techniques. The water-free fabrication of electrosynthesized porous platinum in combination with the possibility to grow it highly localised on 2D or 3D substrates open novel nanotechnology fabrication routes in optoelectronics. The experimental results

from potentiometric measurements have been applied to control the porosity of porous platinum (Supplementary Table 1).

**The influence of the substrate conductivity upon the nanoporous platinum layers**. The infra-red absorbance has been shown to be best for nanoporous platinum layers on copper and silver electrodes[7]. In this work, we studied the mechanism of the formation of nanoporous platinum layers and excluded the influence of electrode material on the process. We used platinum anode, platinum cathode, platinum reference electrode to exclude secondary effects from less noble metals during the electro-deposition. The structure and morphology of nanoporous platinum layers are determined by the electric current density on the cathode, which is a direct function of the electric conductivity of the cathode material. To achieve a similar structure of the porous platinum on different substrates made of flat metallic nanolayers by electrolysis of $PtCl_4$, one has to consider the electric conductivity of the substrate. Platinum has a conductivity of $9.3 \times 10^6$ A/V and silver of $62.1 \times 10^6$ A/V[22]. This means that under identical electrolytic conditions (cathode area and roughness, applied potential, electrolyte concentration, temperature and electrode position) the time needed to build porous platinum of the same thickness on platinum and on silver is by a factor of 6.7 longer for the platinum substrate in comparison to the silver substrate ($62.1 \times 10^6$ A/V/$9.3 \times 10^6$ A/V = 6.7). Namely, porous platinum of similar thickness would be obtained if it grows for 60 s on the platinum substrate and for 9 s on silver (60 s/6.7 = 9 s).

Using the same algorithm of calculation and assuming identical electrolytic conditions, to grow porous platinum of similar thickness for platinum with $9.3 \times 10^6$ A/V conductivity it requires 60 s, for copper with $58.7 \times 10^6$ A/V conductivity it requires 9.5 s, for gold with $44.2 \times 10^6$ A/V conductivity it requires 12.5 s, for aluminium with $36.9 \times 10^6$ A/V conductivity it requires 15.0 s, for molybdenum with $18.7 \times 10^6$ A/V conductivity it requires 30.0 s, for zinc with $16.6 \times 10^6$ A/V conductivity it requires 33.3 s, for nickel with $14.3 \times 10^6$ A/V conductivity it requires 40 s, for palladium with $9.5 \times 10^6$ A/V conductivity it requires 59 s, for steel with $10.1 \times 10^6$ A/V conductivity it requires 54.5 s, for tin with $8.7 \times 10^6$ A/V conductivity it requires 64 s, for lead with $4.7 \times 10^6$ A/V conductivity it requires 120 s, for titanium with $2.4 \times 10^6$ A/V conductivity it requires 240 s, for stainless steel with $1.28 \div 1.37 \times 10^6$ A/V conductivity it requires $408 \div 438$ s, for FeCrAl with $0.74 \times 10^6$ A/V conductivity it requires 750 s.

## Conclusions

Presented work contributes to the understanding of the porous platinum formation on platinum microelectrodes using CV, EIS, XRD, EDX, SEM and HRTEM measurements. The CVs show that there are three separate reduction peaks in aqueous or non-aqueous media: Pt(IV) to Pt(II), Pt(II) to Pt (0) and H(+1) to H (0). The rate of reaction is one order of magnitude larger in aqueous than in non-aqueous media. Hydrogen is involved in the formation of platinum porosity due to its gaseous state. The pore size and the porous platinum layer resistance resulted from non-aqueous electrodeposition, respectively, increase with the applied potential from 25 nm ($Z_{re} = 0.7$ M$\Omega$) at $-0.8$ V to 80 nm ($Z_{re} = 1.6$ M$\Omega$) at $-1.1$ V. For the same concentration of PtCl$_4$, the aqueous electrochemistry produces porous layer with the pore size diameter between 80 and 645 nm in dependence with the additive concentration and applied potential. Meanwhile, the CV and impedance measurement show a clear reaction acceleration event in the presence of the additive, which demonstrates their kinetic role in improving layer adherence. A clear shift of the platinum redox potential was observed in the presence of additive in electrolyte probably due to complexation and ion-pairing or local ionic strength variation. In the potential interval from $-0.8$ to $-1.1$ V, Nyquist plots recoded phase angle from almost 0° to almost 90°, respectively, which means a frequency dependence switch from a chemical reaction-slow electron transfer coupling (0°), to a reversible charge transfer (90°). The reversible charge transfer at the cathode porous platinum-coated proves that the layers are formed and are adherent.

## Methods

**Preparation of the electrolyte**. 99% PtCl$_4$ (MW 336.89 g/mol; Article number CC22008) was supplied by Carbolution Chemicals GmbH, St. Ingbert, Germany. Pb(CH$_3$COO)$_2$ (Article number 1073750250) was supplied by Merck, Darmstadt, Germany. The EDX analysis confirmed the purity of the PtCl$_4$ powder (Fig. 7d). It appears clean except for a very small peak around 1 keV which can be related to the copper L-radiation, which can be a result of secondary fluorescence excitation and must not be an impurity of the PtCl$_4$. The XRD spectra of the PtCl$_4$ powder show instability in light, air and humidity. For electrochemical investigations PtCl$_4$ of 0.05, 0.1, 0.2, 0.3, 0.4, 0.5, 1 and 2% (g/g) solutions in isopropanol with/without 0.01–0.05% Pb(CH$_3$COO)$_2$ were used. The dissolution of Pb(CH$_3$COO)$_2$ in isopropanol takes several hours at room temperature. During this process, the colour of PtCl$_4$ solution in isopropanol turns from brownish to yellow; therefore, the solution is prepared 1 day before the electrolysis. Platinum sputtered container of 1 mL volume (Fig. 6b) serves as the anode, and the cathode was made of platinum wires mounted in the glass (Fig. 6b–e). The disc in contact with the solution has a diameter of 0.25 mm (Fig. 6b, c). The aqueous-based electrochemistry was performed using a similar concentration of PtCl$_4$ solutions prepared in water.

**Electrochemistry**. CV and impedance spectroscopy were performed using a computer-assisted AUTOLAB Metrohm Potentiostat 302N with ADC10M module for high-speed records and FRA32M-module for impedance measurements

equipped with four Platinum-microelectrodes connected to an 8RHD840212 cell sputtered with platinum in the interior, which serves as anode; each electrode disc has a diameter of 0.25 mm (Fig. 6a–e). Nova software was used for data representation.

**Light microscopy**. The micrographs of the electrodes were recorded with a Zeiss Axio light microscope using a 2.5NEOFLUAR objective.

**Scanning electron microscopy (SEM)**. SEM measurements were performed with a field emission microscope JSM-6300F (JEOL, Tokyo, Japan) and FEI Helios NanoLab G3 UC (ThermoFisher, Nederland). The energy of the exciting electrons was mostly 5 keV. In order to enhance the surface sensitivity and in this manner the topographical impression some of the micrographs were taken at a stage tilt of 45°. Besides the detector for secondary electrons (SEI), Everhart-Thornley type, the system is equipped with different detector types (semiconductor and YAG type) for backscattered electrons.

**Focus ion beam transversal cut**. For cutting and immediately imaging the layers, we used FEI-Helios NanoLab G3 UC (Thermo Fisher Scientific, Nederland) dual-beam instrument, which combines a focused ion beam column with a high-resolution field emission (Schottky Thermal Field Emitter) SEM. Retractable detectors for high contrast backscattered electrons BSE imaging are available in this instrument.

**Energy-dispersive X-ray spectrometry (EDX)**. All energy dispersive X-ray analyses were done using a state of the art 30 mm$^2$ silicon drift detector (SDD) by BRUKER (BRUKER Nano GmbH, Berlin, Germany) and the Esprit spectra evaluation software package. The specified energy resolution of the detector at 5.9 keV (Mn-K$\alpha$) is 129 eV.

**X-ray diffraction**. The XRD analysis has been performed with an X′pert Pro Instrument (PANanalytical, Almelo, Netherlands) using Cu-K$\alpha_{1,2}$ radiation. The Scherrer equation was used for the determination of the crystallite sizes.

**Infra-red spectroscopy**. The infra-red spectra of the platinum specimens were measured in an FTIR-Spectrometer (Bruker Instrument). The spectra were recorded with a resolution of 1 cm$^{-1}$ in the spectral range 7500–500 cm$^{-1}$. The samples were illuminated with a tungsten lamp and the spectra were collected with a standard FTIR detector with Mercury Cadmium Telluride (MCT) diode ($D^*$: $>2 \times 10^{10}$ cm Hz$^{1/2}$W$^{-1}$) liquid nitrogen cooled. The measurements were performed at the source aperture, collection mirror velocity and angle that show minimal noise.

**High-resolution transmission electron microscopy (HRTEM)**. The porous platinum film growth on the platinum-coated copper grids was examined using a TEM JEOL NEOARM 200F operating at 200 keV.

**Electrochemical method of the rate constant ($k$) estimation**. The exchange current $I_0$ represents a good indicator of the reaction rate being direct proportional with it[8]. Hereby, one exemplifies the calculation of the standard rate constant[3,8], $k^0$, for the Pt$^{4+}$ reduction described in Eq. (9)

$$Pt^{4+} + 2e \underset{\leftarrow}{\overset{\rightarrow}{}} Pt^{2+} \tag{9}$$

which represents a first-order reaction with the rate, $v$, expressed by the Eq. (10)

$$v = -\frac{d[Pt^{4+}]}{dt} = k[Pt^{4+}] \tag{10}$$

One applies to the Eq. (10) the integration described in Eq. (11) between time 0 s, which corresponds to the initial concentration of Pt$^{4+}$ and time t, which corresponds to the concentration of the Pt$^{4+}$ at the time $t$,

$$\int_0^t \frac{d[Pt^{4+}]}{[Pt^{4+}]} = -\int_0^t k dt \tag{11}$$

to achieve a linear decay dependence of ln[Pt$^{4+}$] with time, indicated in Eq. (12)

$$\ln[Pt^{4+}]\big|_0^t = -kt \tag{12}$$

The exponential form of the Eq. (12) conducts to the relation describes in Eq. (13)

$$\frac{[Pt^{4+}]_t}{[Pt^{4+}]_0} = e^{-kt} \tag{13}$$

To further electrochemically estimate the rate constant $k$ in the Eq. (11), one applies Nernst and Butler-Volmer equations for the equilibrium condition[8].

On one hand, by simply writing the Nernst Eq. (14)

$$E_{eq} = E_{1,0} + \frac{RT}{2F}\ln\frac{[Pt^{4+}]}{[Pt^{2+}]} \tag{14}$$

in the exponential form, we obtain the relation indicated in Eq. (15)

$$e^{\frac{zF}{RT}(E_{eq}-E_{1,0})} = \frac{[Pt^{4+}]}{[Pt^{2+}]} \tag{15}$$

where $[Pt^{4+}]$ and $[Pt^{2+}]$ are the corresponding ion concentrations at equilibrium, $E_{eq}$ (V) and $E_{1,0}$ (V) are the equilibrium and the normal potentials, respectively.

On the other hand, at equilibrium the exchange current, $I_0$ can be expressed by the Butler–Volmer equation[8] for electrode kinetic of $z$ electron transfer

$$I_0 = zFSk^0 [Pt^{4+}]^* e^{-\frac{\alpha zF}{RT}(E_{eq}-E_{1,0})}, \text{ where } [Pt^{4+}]^* - \text{start concentration of } Pt^{4+} \tag{16}$$

Raising the Eq. (15) to the power of – α, one obtains the exponential term of the Eq. (16)

$$e^{-\frac{\alpha zF}{RT}(E_{eq}-E_{1,0})} = \left(\frac{[Pt^{4+}]}{[Pt^{2+}]}\right)^{-\alpha} \tag{17}$$

From Eqs. (16) and (17), the exchange current can be expressed as indicated in Eq. (18)

$$I_0 = zFSk^0 [Pt^{4+}]^{1-\alpha} [Pt^{2+}]^{\alpha} \tag{18}$$

Under the condition that the charge transfer coefficient ($\alpha$) approximates 0.5, at the equilibrium $[Pt^{4+}] = [Pt^{2+}] = C_{eq}$, the Eq. (18) changes to the Eq. (19)

$$I_0 = zFSk^0 C_{eq} \tag{19}$$

$$k^0 = \frac{I_o}{zFC_{eq}S} = \frac{1.08 \times 10^{-6} A}{2 \times 96,485 \, CMol^{-1} \times 49.0625 \times 10^{-5} \, cm^2 \times 7.4 \times 10^{-3} \, Mol \, cm^{-3}}$$

$$= \frac{0.54 \times 10^{-6} C \, s^{-1}}{0.35 \, C \, cm^{-1}} = 1.54 \times 10^{-6} cm \, s^{-1}$$

$k^0$ corresponds to the standard rate constant, Faraday-constant $F = 96{,}485$ C $Mol^{-1}$; gas constant $R = 8.314$ J $Mol^{-1} \, K^{-1}$; absolute temperature $T = 298$ K, the electrode ($\varphi = 0.025$ cm) has the surface area $S = \pi \, r^2 = 49.0625 \times 10^{-5}$ $cm^2$. For 0.5% $PtCl_4$ the corresponding molarity is 14.8 mM. At equilibrium we assumed $[Pt^{4+}] = [Pt^{2+}] = C_{eq} = 7.4$ mM. Eq. (18) can also be directly used for $k^0$ estimation[3].

## Data availability

The datasets generated and analysed during the current study are available from the corresponding authors on reasonable request.

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

## Acknowledgements

We thank A. Dellith for the SEM images and M. Diegel for XRD. Support by the German Research Foundation is gratefully acknowledged (Inst 275/ 391-1).

## Author contributions

J.D. provided SEM, EDX and XRD; A.U. performed HRTEM and FFT; H.S., G.Z., A.I. and MR provided laboratory and materials. O.V. provided the computer-assisted AUTOLAB Metrohm Potentiostat. S.E.S. designed, performed the experiments and wrote the article. All authors read the article and contributed to the discussion of the results. The S.E.S. and H.S. elaborated the revised version according to the reviewer suggestions.

## Funding

## Competing interests

The authors declare no competing interests.
