## [Peer Review File · Communications Chemistry]

Reviewers' comments:

Reviewer #1 (Remarks to the Author):

The authors investigate aspects of the electrochemical formation of porous platinum films both with and without the inclusion of a lead additive, and both in an aqueous and a non-aqueous solution. The studies are of interest for the elucidation of the mechanism of the formation of these porous films, but the study is not yet ready for publication. There are too many unanswered questions to draw the conclusions that the authors are trying to draw.

The authors aim to elucidate the growth mechanism of porous Pt films with and without the inclusion of the Pb additive. The authors should be looking to alternatives to the Pb additive as well though.

The authors also do not have a sufficiently detailed discussion into the mechanism of the formation of the pores within these films and the manuscript in its current form lacks sufficient data analysis and details to support their claims.

The authors state in one place in the manuscript that the addition of the Pb additive led to the disappearance of an oxidation peak, which they attributed to the creation of an irreversible reduction. It is unclear though if the addition of the Pb to the solution did not alter the stability of the Pt oxides, shifting the oxidation potential of the Pt or altering its reduction properties.

Later in the text, the authors state that the Pb additive alters the electrochemical reduction characteristics by shifting the potential at which Pt ions are reduced. They suggest it shifts to higher potentials. Why would the oxidation potential not also shift to higher potentials? The authors should clarify these points. They should also clarify why the Pt salt reduction is shifting in the context of the Pb reduction potentials and the type of structures and compositions that can be formed in the film.

The authors draw the conclusion that the structures of porous Pt layers observed when deposited onto a Cu frame for TEM and STEM analysis are the same as those deposited onto a Pt electrode. The current densities are different, as are the system resistances for the deposition and growth of the Pt layers. It is not clear that these two types of substrates yield the same type of porous layers. Further work is needed to prove the correlation.

The structures formed in the aqueous and non-aqueous solutions are also distinct. Further details are needed to compare each type of structure and their composition. These points are interesting and the comparison is timely for the field, but the study needs to have more correlated data, as well as further discussion into the structures formed under aqueous conditions.

The authors claim that there is an oxidation of the chloride ions observed in their electrochemical data. They need to expand upon their discussion to this regard in further detail. The position of the oxidation potential does change with the scan rate, but it also changes with the addition of the HCl. They seem to rely upon the addition of the HCl to confirm the presence of this peak and its position, but there are further complications to their charge transport resistances and to the composition of the double layer and IHL and OHL, as well as the fact that the peak positions they observe don't match with those they are trying to assign in other CVs. They will need to carefully discuss these factors in the manuscript.

The units provided need to be clarified in the manuscript, such as the levels of % species is unclear. Is this a weight/weight percentage? or weight to volume percentage? It is difficult to determine the exact concentration from these units without clarity.

The effect of bubble formation is eluded to, but the discussion on hydrogen bubble formation and its correlation with the porous layers is not discussed in detail. There is much prior art on the use of hydrogen bubble templating. The authors should also carefully consider these prior works in the context of their own work.

The discussion of pore size should also be included in the author's discussion of the mechanism by which these porous layers are formed. They need to discuss in further detail why they believe, based on the data, that the pores of these specific dimensions (e.g., 25 and 35 nm) are formed in their studies.

The authors suggest in their schematics that cubic materials are being formed in the process of their electrodeposition, but that is unclear from the data presented.

The EDX analysis does not account for the potential inclusion of Pb into their porous layers. The signal could be overlapping with the assigned peaks. Further analysis and additional elemental analysis techniques are needed to understand what happens to the Pb added into the solution. Is it formed as an inclusion? Is it within the outermost layers of Pt, impeding the formation of the Pt oxides? How do they know?

The IR reflectance spectra do not have as low an intensity as those in their previous reports (Scientific Reports volume 7, Article number: 14955 (2017)). The authors should address this observed difference.

The authors add up to 0.05% of the Pb additive. This appears to be in excess as there is a predominance of other structures formed under these conditions (e.g., see SEM data in Figure 6). The same features are observed at 0.03%. It appears that the Pb is deposited onto these surfaces, but could be segregating from the Pt that is deposited when the concentrations are relatively high. The authors need a more careful analysis of the composition and structure of the materials formed at each concentration and reaction condition (e.g., aqueous and non-aqueous).

The authors suggest that the capacitance from adsorbed species is described in Figure 8A, but it is not. Further revisions are necessary (See figure 8 caption.).

The impedance plots offer a lot of information. The authors should analyze this data in more detail, such as the correlation between features therein and the structures being formed.

Many of the labels in the figures (e.g., legend in Figure 6) are too small to properly read. The authors need to redo the layout of their figures so that the data can be properly assessed in detail.

There are a number of grammatical mistakes that need to be addressed for clarity in the meaning of the authors writing to assist in understanding their meaning, such as:

"The CVs give insides"... incorrect terminology

"charge of the inner Helmholtz layer (IHL), the charge of the inner Helmholtz layer (IHL), ..." repeats

"which indicates a pregnant involvement" ... unclear what was meant here

"this reasons the unclosed"...

"amounts 129 eV..."

"isopropanolat" ...

Reviewer #2 (Remarks to the Author):

The authors report a set of experiments for porous platinum nanolayers. They find that the platinum cation's electrochemical reduction includes three steps: two steps for Pt and one step relates to the hydrogen reduction. The experimental efforts employ a set of electrochemistry approach and characterize the resulting porous Pt nanolayers. The main conclusion the authors draw from their study is the reduction process of the platinum cation to form porous platinum nanolayers with and without additives.

The electrochemical experiments are properly organized and well interpreted. The authors argue that the pore size and thin film growth kinetics are related to the applied potential and added additive. The work is self-supportive and interesting for the mechanism understanding of the aqueous electrochemical synthesis.

However, to what extent the influence of the applied potential has on the porosity is not clear. Porosity should be calculated and given values. No practical controlment or application of the porosity based on the experiment interpretation is demonstrated in this work. Besides, FFT for TEM images in Figure 1 is not indexed. Figures 6 and 7 are referred to in the main text in the reversed order. Figure 7 is mislabeled.

Reviewer #3 (Remarks to the Author):

The paper investigated the deposition of Pt porous layers and the underlying growth mechanism and kinetics. The result is interesting and useful. I recommend it for consideration in Communication Chemistry after minor revision. Additional comments list as follows:

- (1) From XRD and HRTEM image in Fig. 1D and 2E, why is the crystallinity of Pt layer so poor, as we known, electrodeposition is an effective method to produce high-quality films?
- (2) From FFT in Fig. 2E, the paper demonstrated the Pt nanoparticles is single crystalline. More data should be provided.
- (3) The size of Pt nanoparticles from SEM (Fig. 1B) and HRTEM (Fig. 1E) is not matched.

Manuscript title "Electrochemical Growth Mechanism of Nanoporous Platinum Layers"

We thank the reviewers for their constructive criticism. Corresponding parts of the manuscript were revised and results from additional SEM, HRTEM, and FFT investigations have been included. The point-by-point answer to the reviewer questions are presented below.

Reviewer #1:

The authors investigate aspects of the electrochemical formation of porous platinum films both with and without the inclusion of a lead additive, and both in an aqueous and a non-aqueous solution. The studies are of interest for the elucidation of the mechanism of the formation of these porous films, but the study is not yet ready for publication. There are too many unanswered questions to draw the conclusions that the authors are trying to draw.

The authors aim to elucidate the growth mechanism of porous Pt films with and without the inclusion of the Pb additive. The authors should be looking to alternatives to the Pb additive as well though.

Answer: We also tested copper acetate, copper sulphate and acetic acid as additives in aqueous and non-aqueous electrochemical deposition of porous platinum. We observed that the adherence of the porous Pt layer on different substrates (Cu, Pt, In,..) is better if lead acetate additives are used instead of Cu acetate. This is in agreement with results from Kurlbaum and Lummer.

F. Kurlbaum and O. Lummer (Verh. Phys. Ges. Berlin, 14, 56,1895) used copper and lead salts to increase the adherence of porous platinum on the substrate. They also obtained better results using lead acetate as additive instead of copper sulphate.

Feltham and Spiro (Chem. Rev. 71, 177-193 (1971) also reported on the increased adherence of platinum black in the presence of lead acetate as an additive.

We motivate the use of lead acetate additives in the revised manuscript as follows: "We present here the adherent porous platinum layer of controlled thickness and porosity electrosynthesized in aqueous and non-aqueous media in the presence of lead acetate which has shown to be an additive effectively supporting adherence of Pt layers on different substrates. Kurlbaum and Lummer used copper sulphate and lead acetate to increase the adherence of porous platinum on the substrate, however they obtained better results using lead acetate¹. Feltham and Spiro³ also reported on good adherence of platinum black in the presence of lead acetate. We have tested copper acetate, copper sulphate, and acetic acid as additives in the aqueous and non-aqueous electrochemical baths. We also observed that the adherence of the porous platinum layer on different substrates (Cu, Pt, Al, In-Sn, Ag, Au) is better if lead acetate additives are used instead of copper salts".

The authors also do not have a sufficiently detailed discussion into the mechanism of the formation of the pores within these films and the manuscript in its current form lacks sufficient data analysis and details to support their claims.

Answer: In the revised paper we included additional HRTEM images to discuss formation of a crystalline initialization Pt layer before nanoporous Pt layer is formed on top. We added additional HRTEM images and corresponding discussion.

The authors state in one place in the manuscript that the addition of the Pb additive led to the disappearance of an oxidation peak, which they attributed to the creation of an irreversible reduction. It is unclear though if the addition of the Pb to the solution did not alter the stability of the Pt oxides, shifting the oxidation potential of the Pt or altering its reduction properties.

Answer: We thank the reviewer for hinting out a possible explanation and included his/her explanation into the revised manuscript as follows: "The stair case CV wave of oxidation in the interval of 0.8 V ÷ -0.5 V, detectable at a scan rate of 0.25V/s, which is attributed to the platinum oxidation, is no longer observable in the system containing additive (Fig. 6G), which implies that it has undertaken an irreversible reduction or that the additive alters the stability of the oxidized platinum species, shifting the oxidation potential to higher values".

Later in the text, the authors state that the Pb additive alters the electrochemical reduction characteristics by shifting the potential at which Pt ions are reduced. They suggest it shifts to higher potentials. Why would the oxidation potential not also shift to higher potentials? The authors should clarify these points.

Answer: The shift of the reduction potential at which Pt ions are reduced to higher values in the presence of additives is discussed in detail in revised manuscript. However, shift of oxidation potential at which Pt ions are oxidized to high values in the presence of additives can not be clearly discussed because of the superimposed CV signals from oxidations of chloride and electrolyte. We take the reviewer suggestion of possible oxidation peak shift and included into the manuscript" in the presence of additive the oxidation wave in the interval of 0.8 V ÷ -0.5 V is no longer observable in the system containing additive (Fig. 6G), which implies that it has undertaken an irreversible reduction or that the additive alters the stability of the platinum ion species, shifting the oxidation potential to higher values."

The revised manuscript contains the following interpretations of the cyclic voltammograms:

Figure. 1 for reviewer 1. Cyclic voltammetry study of PtCl_4 in isopropanol; (A-B) CVs of 0.05% PtCl_4 in isopropanol recorded at 0.1 V/s in the absence of 0.01% $\text{Pb}(\text{CH}_3\text{COO})_2$ (A) and in the presence of 0.01% $\text{Pb}(\text{CH}_3\text{COO})_2$ (additive)(B). CV starts at +2 V(A-B). For $\text{Pt}^{4+} + 2e \rightarrow \text{Pt}^{2+}$, $E_1 = +0.10$ V (without additives), and $E_1^* = +0.25$ V (with additives), for $\text{Pt}^{2+} + 2e \rightarrow \text{Pt}^0$; $E_2 = -0.4$ V (without additives), and $E_2^* = -0.2$ V (with additives); C, The cartoon of the standard free energies during the platinum ion reduction activation. Standard free energy for the reactant Pt^{4+} , ΔG_R^* , $(\Delta G_R^*)_{\text{Pb}}$, and for the product $\text{Pt}^{(4-z)+}$, ΔG_P^* , $(\Delta G_P^*)_{\text{Pb}}$ in the absence and presence of lead, and the energy amount, $zF\Delta E$, with which the reaction curve is lowered by lead are indicated on the panel. G free energy (kJ/mol); ΔG_R^* standard free energy of activation ($\text{J}\cdot\text{Mol}^{-1}$); ΔG_P^* standard free energy of activation for product ($\text{J}\cdot\text{Mol}^{-1}$); ΔG_P^* standard free energy of activation for reactant ($\text{J}\cdot\text{Mol}^{-1}$); E electrochemical potential (V), difference between Galvani potential of metal ϕ^M and of solution ϕ^S ; z number of electron transfer; F Faraday number $F=96485$ C $\cdot\text{Mol}^{-1}$; $zF\Delta E$ -Energy of electrons on electrode which is changed due to potential change ($\text{J}\cdot\text{Mol}^{-1}$);

Cyclic voltammetry study of PtCl₄ in isopropanol. “The system in Fig. 1A does not contain additive and does not generate adherent porous platinum. In contrast, the investigated system in the Fig. 1B contains additive and generates adherent platinum black. We observed that the additive considerably changes the CV shape, even if the additive alone does not show prominent electroactivity at the used concentration. A small reduction wave starting at -0.2 V and an oxidation wave starting at 0.6 V can be identified on the I-E curve. We recorded a shift of the platinum ion reduction peaks at a more positive potential of approximate 0.2 V (Fig. 1B) and an increase in current intensities due to the additive kinetic involvement (Fig. 1B). The lead additive is not observed in the XRD diagram of platinum deposits, which implies that platinum does not concurrently grow with lead. The additive can also inhibit the hydrogen formation and increase the coulometric efficiency of platinum layer growth¹². In Fig. 1B, four reduction waves are identified on the CVs, which were shifted towards the positive potential compared to those in Fig. 1A. The electrode surface might change because the deposition causes deviations in the platinum ion reduction peak (the peaks of the black, dark blue, blue and light blue waves appear at -0.7 V, -0.65V, -0.6V and -0.55V, respectively, in Fig. 1B). The following hypothesis is true but not clearly sustainable: at the cathode, the direct four-electron reduction of platinum ions (Pt⁴⁺ +4e⁻ →Pt↓) with gaseous hydrogen formation is more probable for the first system (Fig. 1A) than for the second one (Fig. 1B). A metastable four - electron transfer of platinum oxido-reduction centred at -0.5 V was identified ...; however, with the addition of PtCl₄, the wave of four-electron oxidation splits into two waves of two - electron oxidation. The CVs in Fig. 1A supports a two-step mechanism: Pt⁴⁺ + 2e⁻ → Pt²⁺ and Pt²⁺ + 2e⁻ →Pt↓. In Fig. 1A, the evolution of the hydrogen reduction peak is visible between -0.75 V and -1.5 V, but this peak is absent in Fig. 1B. Here, the peak is replaced by a tail of reduction in this interval, which can be attributed to the hydrogen ion reduction. The wave of oxidation in the interval of 0.8 V ÷ -0.5 V assigned to the platinum oxidation, is no longer observable in the system containing additive (Fig. 6B), which suggests that platinum ions have undertaken an irreversible reduction or that the oxidation of platinum ion species shifts to higher potential values. To resume our findings, the CVs in the presence of the additive shows three separate reductions as follows: 1) Pt(IV) +2e⁻→ Pt(II) (standard rate constant 1.5x10⁻⁶ cm s⁻¹) begins at -0.17 V, 2) Pt(II) +2e⁻→ Pt (0) (standard rate constant 3.2x10⁻⁶ cm s⁻¹) begins at -0.3 V, and 3) hydrogen reduction (2H⁺+2e⁻→H₂↑) begins at -0.75V. Hydrogen acts in the formation of platinum porosity by the gaseous state and not by the reductive effect. We attribute the role of porosity generator to the hydrogen gas.

To summarize, cyclic voltammogram data at the reduction potential of platinum ions in aqueous and non-aqueous medium (Figs. 6,7) confirm that the reduction potential of platinum without additives lies at more negative values than the reduction potential of platinum with additives. For the first step of reduction Pt⁴⁺ +2e⁻→Pt²⁺ the electrochemical potentials are E₁=+0.10 V (without additives), and E₁^{*}=+0.25 V (with additives). The resulting potential difference ΔE_{1,1}^{*}=E₁- E₁^{*} = -0.15 V reduces the standard free energy (Fig. 6H). For the second step of reduction Pt²⁺ +2e⁻→Pt⁰ ; the electrochemical potentials are E₂= -0.4 V (without additives) and E₂^{*}= -0.2 V (with additives). Also here the resulting potential difference ΔE_{2,2}^{*}=E₂-E₂^{*} = -0.2 V reduces the standard free energy (Fig. 6H). The reduction of free energy is lowered and amounts to -zFΔE_{1,1}^{*}=28.9kJ/mol and to -zFΔE_{2,2}^{*}=38.6 kJ/mol for the first and second step, respectively.”

Figure 2 for reviewer1. Evolution of PtCl₄ CVs in water. A, CVs of aqueous 0.5% PtCl₄ (black line) and 0.01%Pb(CH₃COO)₂ (grey line); B, CVs of aqueous 0.5% PtCl₄ , 0.01% Pb(CH₃COO)₂ (nuanced blue lines) compared to 0.01%Pb(CH₃COO)₂ (black line);

Cyclic voltammetry study of PtCl₄ in water. “ The stair case CV of PtCl₄ 0.5% in water is plotted in figure 2A (black line) and shows a prominent peak of platinum reduction, one peak of hydrogen ions reduction at -1 V and two oxidation shoulders: the oxygen ion oxidation partially overlapped with the chloride oxidation (+1V). The presence of the additive (Fig. 2B) generates a completely changed aqueous CV with four well-defined peaks of oxidation and four peaks of reductions, which show different kinetics and a clear separation between hydrogen reduction and platinum ion reduction. Using the procedure from Feltham et al.³, we estimated the rate constant of the first-order reactions: Pt(IV) +2e→ Pt(II) at a standard rate constant of 1.4x10⁻⁵cm s⁻¹and Pt(II) +2e→ Pt (0) at the standard rate constant of 2.2x10⁻⁵cm s⁻¹. The corresponding reduction potentials are more negative than those reported by Feltham et al.³

For the aqueous electrochemistry, the CVs data show a clear electro-kinetic difference between PtCl₄/water (no platinum porous) and PtCl₄/water/additive (platinum porous) (Fig. 8). The additive shifts the redox potential of platinum ions to more positive potentials and increases the platinum ion reduction rate. For the first step of reduction Pt⁴⁺ +2e→Pt²⁺ the electrochemical potentials are E₁=+0.05 V (without additives), and E₁*=+0.35 V (with additives) resulting a potential difference ΔE_{1,1}* =E₁- E₁* = -0.30 V. For the second step of reduction Pt²⁺ +2e→Pt⁰ ; E₂= -0.4 V (without additives), and E₂*= -0.25 V (with additives), the potential difference is ΔE_{2,2}* =E₂-E₂* = -0.15 V. This lowers the reaction curves, respectively the free activations energies with -zFΔE_{1,1}*=57.9kJ/mol for the first step and with -zFΔE_{2,2}*=28.9 kJ/mol for the second step.

They should also clarify why the Pt salt reduction is shifting in the context/in the presence of the Pb reduction potentials and the type of structures and compositions that can be formed in the film.

We related the dependence of the inner potential profile on the distance of ions from the platinum cathode (Fig. 12 F middle) with the formation of electric double layer. On one

hand, the IHL part of the electric double layer with additives (Fig. 12F top) is larger than the IHL part of the electric double layer without additives (Fig. 12F bottom). This is related with the larger ionic radius of the Pb^{2+} additives in comparison to the ionic radius of Pt^{4+} . On the other hand, the cyclic voltammograms show that the additive moves the electrochemical potential of platinum oxido-reduction to the right resulting $\Delta E = E(\text{additive}) - E(\text{without additive}) > 0$. From this we conclude that the potential of the platinum ions in the solution phase of the case with additive is less negative than the potential of the platinum ions in the solution phase of the case without additives. Therefore, the activation energy of the solution without additives is larger than the activation energy of the solution with additives (Fig. 6H). For the reaction $Pt^{4+} + 2e^- \rightarrow Pt^{2+}$ and $Pt^{2+} + 2e^- \rightarrow Pt^0$, without additives the corresponding potential of the reaction (Fig. 6F) lies at $E_1 = +0.10$ V (without additives) and at $E_2 = -0.4$ V (without additives). For the reaction $Pt^{4+} + 2e^- \rightarrow Pt^{2+}$ and $Pt^{2+} + 2e^- \rightarrow Pt^0$, with additives the corresponding potential of the reaction (Fig. 6G) lies at $E_1^* = -0.15$ V and at $E_2^* = -0.2$ V, resulting a potential difference $\Delta E_{1,1}^* = E_1 - E_1^* = -0.15$ V for the first step of reduction and $\Delta E_{2,2}^* = E_2 - E_2^* = -0.2$ V, for the second step of reduction. This lowers the reaction curves, respectively the free activations energies with $-zF\Delta E_{1,1}^* = 28.9$ kJ/mol for the first step and with $-zF\Delta E_{2,2}^* = 38.6$ kJ/mol for the second step. Finally, one can conclude that the observed shift of the potential to the right is related with additives having a larger radius than the replaced ions.

Figure 3 for reviewer 1, Diagram of electric double layer formation at the electrode in the absence (lower part) and presence (upper part) of lead. The corresponding electrical potential profile (middle part) across the electrode electric double layer (ϕ^M ; ϕ^{IHL} ; ϕ^{OHL} ; ϕ^d ; ϕ^S) built from the charge of inner Helmholtz layer (IHL) (σ^{IHL}), charge of outer Helmholtz layer (OHL) (σ^{OHL}), and charge of diffuse layer (σ^d); Blue and black lines are used to plot the electrical potential with and without additives, respectively. ϕ^M -inner potential (Galvani potential) of species in the metal phase (V) ϕ^S -inner potential (Galvani potential) of species in the solution phase (V); ϕ^{IHL} potential of inner Helmholtz layer (V); ϕ^{OHL} potential of outer Helmholtz layer (V)

We included in the revised manuscript a table summarizing the pore size as a function of applied potential, the additive concentration and electrolyte nature (aqueous/non-aqueous). We also included an explanation of shifting the platinum oxido-reduction potential by the addition of $Pb_2(CH_3COO)_2$ on the basis of moving the electrical equilibrium, as follows:

“In both cases, aqueous and nonaqueous, lead ions cause a shift of the redox potential to smaller values. This shift can be related with the larger ionic radius of the Pb^{2+} ($r=119pm$)²¹ additives in comparison to the ionic radius of the replaced Pt^{4+} ($r=62.5pm$)²¹ ions and to the fact that lead ions make local changes related to complexation, ion pairing or ionic strength variation.

To correlate the electrode potential shift with the thermodynamic properties of the system in the presence of lead, we start from the definition of electrochemical potential.

The electrochemical potential at the interface enables the platinum reduction:

This electrode process involves a fast charge-transfer governed by the Nernst equation.

By definition, the electrochemical potential, for species Pt with the charge Z and inner potential ϕ in phase M, $\overline{\mu}_{Pt}^M$, respectively in the phase S, $\overline{\mu}_{Pt}^S$, is defined by the equation (2):

$$\overline{\mu}_{Pt}^M = \mu_{Pt}^M + z F\phi^M \quad \text{and} \quad \overline{\mu}_{Pt}^S = \mu_{Pt}^S + z F\phi^S \quad (2)$$

In the equation 2, the chemical potential for the same species in phase M, μ_{Pt}^M , respectively, in phase S, μ_{Pt}^S , has the correspondence to thermodynamic parameters:

$$\mu_{Pt}^M = \left(\frac{\delta G}{\delta n_{Pt}} \right)_{T,P} = \left(\frac{\delta H - T\delta S}{\delta n_{Pt}} \right)_{T,P} \quad (3)$$

$$\mu_{Pt}^S = \left(\frac{\delta G}{\delta n_{Pt}} \right)_{T,P} = \left(\frac{\delta H - T\delta S}{\delta n_{Pt}} \right)_{T,P} \quad (4)$$

where n_{Pt} is the number of moles of Pt^{Z+} in the phase M or S; G,H,S are the thermodynamic parameters: free energy, enthalpy and entropy, respectively.

At interphase (M/S) equilibrium (eq), the equality between the electrochemical potentials of the species Pt^{Z+} in the two phases is achieved:

$$\overline{\mu}_{Pt}^M = \overline{\mu}_{Pt}^S \quad (5)$$

By combining and rearranging the eq. 3, eq. 4 and eq. 6, the following relationship can be written:

$$\left[\left(\frac{\delta H - T\delta S}{\delta n_{Pt}} \right)_{T,P}^M + z F\phi^M \right]_{eq} = \left[\left(\frac{\delta H - T\delta S}{\delta n_{Pt}} \right)_{T,P}^S + z F\phi^S \right]_{eq} \quad (6)$$

$$\left[\left(\frac{\delta H - T\delta S}{\delta n_{Pt}} \right)_{T,P}^S - \left(\frac{\delta H - T\delta S}{\delta n_{Pt}} \right)_{T,P}^M \right]_{eq} = [z F(\phi^M - \phi^S)]_{eq} = zFE_{eq} \quad (7)$$

chemical term

electrical term

where the Galvani potential difference between the metal/solution phases (Eq.8) expresses the electrode potential (E); $zF(\phi^M - \phi^S)$ in the Eq. 7 represents the electrical component of the free energy (G), respectively of the work/energy necessary to transfer z electrons across the metal/solution interface¹⁸. The left part of equation 7 shows how the thermodynamic parameter change (H, S) and influence the redox potential. In this context, Pb modifies the interphase thermodynamic equilibrium and charge equilibrium. On one hand, the Pb ions cause changes in the interfacial potential difference by altering the charge balance and the charge density at the interface, due to a larger ionic radius (119pm compared with 62.5pm for Pt^{4+})²¹. On the other hand, the electrochemical potential of each phase depends on the

associated enthalpy, entropy. Therefore, the interfacial potential differences can also occur without charge excess at interface as described by Eqs. 2-7. The interfacial potential difference, E (Eq. 7) determines the electro-kinetic (fast charge transfer) event at the electrode, being complexly influenced by the thermodynamic parameters at the boundary metal-solution. Fig. 6H shows the effect of potential change on the standard free energies during the platinum reduction (Eq. 1) caused by the lead additive. Lead moves the electrochemical potential to more positive values with ΔE in both aqueous and non-aqueous medium. The relative energy of the electrons on the cathode also changes with $-zF\Delta E$ and the reaction curve moves down with this amount of energy, with which the activation energy of the reaction is lowered. It is realistic to accept that lead ions modify the interfacial entropy and enthalpy (increase in enthalpy due to the binding forces increase) with consequences upon the interfacial electrical equilibrium.“

Why would the oxidation potential not also shift to higher potentials?

Answer: We were focused on the platinum ion reduction and stressed the irreversible reduction of platinum ions because it is related to the platinum deposition; however, we do not exclude the shift of oxidation potential of the platinum ions.

The authors draw the conclusion that the structures of porous Pt layers observed when deposited onto a Cu frame for TEM and STEM analysis are the same as those deposited onto a Pt electrode. The current densities are different, as are the system resistances for the deposition and growth of the Pt layers. It is not clear that these two types of substrates yield the same type of porous layers. Further work is needed to prove the correlation.

Answer: We completely agree with this comment from the reviewer and repeated growth of Pt layers on Pt for HRTEM analysis. Previous where Pt layers on Cu have been analysed by HRTEM are not comparable because a Pt layer in $PtCl_4$ solution will grow on Cu without applying an electrical potential. This is in contrast to the growth of Pt layer on Pt where an electrical potential has to be used. For the additional HRTEM analysis we used a Pt-coated copper grid instead of the previously used uncoated Cu grid for HRTEM images.

HRTEM (Fig. 1E): In the old manuscript we used a Cu grid with initiation porous Pt layer for HRTEM imaging. However, the quality of HRTEM images was poor. This might be related with the dependence of the porosity of electrodeposited porous Pt layers on the material of underlying electrode. In revised paper, we prepared porous Pt layers on a Pt coated grid. Now, as expected, the porosity of electrodeposited porous Pt layers on Pt coated grids and on high-quality Pt layers is comparable, e.g. SEM images. In the revised manuscript HRTEM and FFT data from porous Pt layers on a Pt coated grid are provided to demonstrate the crystallinity of porous Pt layers.

We thank the reviewer for stressing this point.

Figure 4 for reviewer 1. Porous platinum layer seen in transmission electron microscopy. A-N, TEM images of the porous platinum growth on a platinum coated copper grid (A,B,C,D,F,H,M) and the Fast Fourier Transform, FFT, (E,G,J,K,L,N) of each indicated areas (I,II,III,IV,V,VI).

HRTEM were recorded at the margins of the porous layer growth (Figure 4 for reviewer) on platinum coated copper grid. TEM images show several types of particles at the margins of the layer, some of them are amorphous and some are crystalline in the core with an amorphous shell. The FFT shows that crystalline particles are not single crystals, they contain defects (i.e. twins). The shape of the amorphous particles tends to be round, crystalline ones tend to be elongated. It might be possible that amorphous and crystalline particles form simultaneously.

The structures formed in the aqueous and non-aqueous solutions are also distinct. Further details are needed to compare each type of structure and their composition. These points are interesting and the comparison is timely for the field, but the study needs to have more correlated data, as well as further discussion into the structures formed under aqueous conditions.

Answer: Yes, differences in reduction kinetics and porosity are huge. The process of Pt reduction in aqueous solution is one order of magnitude faster in comparison to the Pt reduction in non-aqueous solutions. Furthermore, hydrogen is abundant in aqueous solutions because the dissociation of water is larger in comparison to the dissociation of isopropanol. The Pt structure is mainly determined by porosity. The porosity in aqueous solution is in the μm range (SEM images in Fig. 9C-F) in comparison to the porosity in non-aqueous solution which is in the nm range (SEM image in Figure 5).

The authors claim that there is an oxidation of the chloride ions observed in their electrochemical data. They need to expand upon their discussion to this regard in further detail. The position of the oxidation potential does change with the scan rate, but it also

changes with the addition of the HCl. They seem to rely upon the addition of the HCl to confirm the presence of this peak and its position, but there are further complications to their charge transport resistances and to the composition of the double layer and IHL and OHL, as well as the fact that the peak positions they observe don't match with those they are trying to assign in other CVs. They will need to carefully discuss these factors in the manuscript.

Answer: The Cl⁻ oxidation peak has been assigned by adding HCl of different concentration. Corresponding CV experiments have been performed to confirm Cl⁻ oxidation peak. As pointed out by the reviewer there are further details related with HCl. We decided to keep labeling of Cl⁻ oxidation peak where it is visible, but to remove the CV data which belong to the experiments where HCl has been added.

The units provided need to be clarified in the manuscript, such as the levels of % species is unclear. Is this a weight/weight percentage? or weight to volume percentage? It is difficult to determine the exact concentration from these units without clarity.

Answer: The missing units have been added in the revised manuscript.

The effect of bubble formation is eluded to, but the discussion on hydrogen bubble formation and its correlation with the porous layers is not discussed in detail. There is much prior art on the use of hydrogen bubble templating. The authors should also carefully consider these prior works in the context of their own work.

Answer: In revised manuscript we discussed the bubble formation in more detail and added prior art.

“To create porous electrode, the co-generation of hydrogen along with metal deposition, represents a scientific interest for long time¹⁻³. The co-generation of hydrogen takes place at the electrochemical generation of platinum black at the cathode in aqueous media¹⁻³ and in non-aqueous media⁷. The role of hydrogen abundance at the cathode during the metal deposition is directly connected with the porosity of the metal foam formation. There is a recent review which compiles the factors (electrolyte composition, temperature, applied potential) influencing foam formation using the hydrogen bubble templating method¹⁸. Aligned to this idea, that the abundance of hydrogen increases at the more negative potential (i.e. from -0.7V to -1V), we observed an accentuated dependence of the porosity on the applied potential with all other electrolysis conditions being identical: at -0.8 V, the crystallite size was 20 nm; at -0.9 V, it became 35 nm; at -1 V, it was 50 nm (Fig. 5). The pore sizes of the sponge platinum in aqueous electrochemistry are 100 times larger and can reach 300 nm (Figs. 9C-F).”

As an additional reference we included:

18. B. J. Plowman, L. A. Jones, S. K. Bhargava, Chem. Commun. 2015, 51, 4331–4346.

We observed and stated in the Table 1 that the porosity of platinum layer increases in both aqueous and non-aqueous media at the more negative potentials. A plausible explanation to this phenomena can be that the increase of hydrogen evolution at more negative potential increases the entropy of the system and the structures are less compact.

The discussion of pore size should also be included in the author's discussion of the mechanism by which these porous layers are formed. They need to discuss in further detail

why they believe, based on the data, that the pores of these specific dimensions (e.g., 25 and 35 nm) are formed in their studies.

Answer: Based on new HRTEM data (Fig. 2) we discuss the nanoporosity of Pt grown in non-aqueous solutions. We also added discussion on porosity of Pt grown in aqueous solutions.

The authors suggest in their schematics that cubic materials are being formed in the process of their electrodeposition, but that is unclear from the data presented.

Answer: XRD data reveals cubic structure of Pt with a lattice constant of $a=3.9231\text{\AA}$. This agrees with cubic Pt from XRD database.

The EDX analysis does not account for the potential inclusion of Pb into their porous layers. The signal could be overlapping with the assigned peaks. Further analysis and additional elemental analysis techniques are needed to understand what happens to the Pb added into the solution. Is it formed as an inclusion? Is it within the outermost layers of Pt, impeding the formation of the Pt oxides? How do they know?

Answer: Yes, we assume that Pb is included in μ pores of Pt layers grown in aqueous solution at high concentration of Pb. "EDX does not show the presence of lead in the porous platinum layer obtained in non-aqueous media. However, the presence of lead in platinum porous layer obtained in aqueous media at high concentration (at and above 0.03%) of lead acetate is indicated by EDX and SEM (Fig. 9E,F). This difference in the Pb inclusion can be related with the larger pore size of Pt layer obtained in aqueous solution in comparison to Pt layers obtained in non-aqueous solution."

The IR reflectance spectra do not have as low an intensity as those in their previous reports (Scientific Reports volume 7, Article number: 14955 (2017)). The authors should address this observed difference.

Answer:

A

Figure 6. Optical properties of the *in situ* electro-assembled platinum black thin layer on different substrates. The FTIR reflectance spectra referred to silver mirror of the platinum black layers achieved in 90 s in non-aqueous media on ITO (black line), tin-copper alloy (red line), aluminium (blue cyan), copper (grey), gold (olive), platinum (green), silver (magenta) (the other electrolysis conditions described in \$Methods); the insets illustrate the photographs of the platinum black on different substrates indicated on the panels.

B

Figure 2 Backscattering electron microscopy reveals platinum crystals inside the layer: A, SE (A) and BSE(B) SEMs of the platinum assembly; C, Drawing of the layer; D, EDX diagram of porous platinum film on platinum cathode; E, XRD diagram and the difference XRD spectra of porous platinum film growth on platinum and platinum cathode; F, FTIR reflectance spectra in the wavenumber region 7500cm^{-1} - 500cm^{-1} of the platinum porous at 250s electrolysis (blue lines) compared with the bare platinum electrode (green) and Al mirror (black line).

Figure 5 for reviewer 1. (A) Fig. 6 of Scientific Reports volume 7, Article number: 14955 (2017) we only show IR reflectance spectra of Pt black on Pt electrode (green line); (B) Figure 2 of the original manuscript.

In Fig. 6 of Scientific Reports volume 7, Article number: 14955 (2017) we only show IR reflectance spectra of Pt black on Pt electrode (green line in Fig. 6). In Fig. 2 of present manuscript we show IR reflectance spectra of Pt electrode (green line in Fig. 2f) and of Pt black on Pt electrode (blue line in Fig. 2f). The intensity of IR reflectance spectra of Pt black

on Pt electrode has same small value of IR reflectance. To explain difference in IR reflectance spectra of Pt black and porous Pt on Pt black, we revised discussion of Fig. 2f as follows: “Fourier Transform Infrared (FTIR) spectra recorded for the cathode (Fig 4E), before (green line), and after platinum deposition (blue line) show low reflectance of the porous platinum layer in the wavenumber region $500\text{cm}^{-1} \div 7500\text{cm}^{-1}$ in agreement with our previous reported data⁷”.

The authors add up to 0.05% of the Pb additive. This appears to be in excess as there is a predominance of other structures formed under these conditions (e.g., see SEM data in Figure 6). The same features are observed at 0.03%. It appears that the Pb is deposited onto these surfaces, but could be segregating from the Pt that is deposited when the concentrations are relatively high. The authors need a more careful analysis of the composition and structure of the materials formed at each concentration and reaction condition (e.g., aqueous and non-aqueous).

Answer: Presence of Pb at high concentration (at and above 0.03%) is visible in EDX and SEM spectra and is discussed in revised manuscript as follows: “EDX does not show the presence of lead in the porous platinum layer obtained in non-aqueous media. However, the presence of lead in platinum porous layer obtained in aqueous media at high concentration (at and above 0.03%) of lead acetate is indicated by EDX and SEM (Fig. 9E,F)”

The authors suggest that the capacitance from adsorbed species is described in Figure 8A, but it is not. Further revisions are necessary (See figure 8 caption.).

Answer: We thank the reviewer for pointing out this error. We corrected it. It is Fig. 8B where the capacitance of the adsorbed Pt 4+ (Cads) is introduced and illustrated in more detail in Fig. 8F.

The impedance plots offer a lot of information. The authors should analyze this data in more detail, such as the correlation between features therein and the structures being formed.

Answer: In the revised manuscript subfigures C, D, G, H have been removed. Detailed analysis of those figures includes modelling of equivalent circuit. Extraction of model parameters RW , Cd , Rct , $Cads$, C , R , $CpPt$ is ongoing and will be discussed in another work. For this manuscript we decided to keep Fig. 9 A,B, D, F only. Just by comparing the impedance data as a function of time one can recognize the changes in imaginary and real impedances. Fig. 11 shows that the Z_{re} values recorded at -0.7V and -1V without additive, (Fig. 11A) exhibit a decrease compared to the case with additives (Fig. C) after 150 s. This difference occurs because of insufficient adherence. As a consequence, the electrode remains void of layers and displays higher electrical conductivity and lower impedance. Z_{im} constantly increases with the more negative applied potential at times shorter than 130 s (Figs. 11B,D). This capacitive increase is slightly more accentuated in the presence of additives (Figs. 11B,D). At longer times than 150 s, Z_{im} decreases in both cases.

Reviewer #2 (Remarks to the Author):

The authors report a set of experiments for porous platinum nanolayers. They find that the platinum cation's electrochemical reduction includes three steps: two steps for Pt and one step relates to the hydrogen reduction. The experimental efforts employ a set of electrochemistry approach and characterize the resulting porous Pt nanolayers. The main conclusion the authors draw from their study is the reduction process of the platinum cation to form porous platinum nanolayers with and without additives.

The electrochemical experiments are properly organized and well interpreted. The authors argue that the pore size and thin film growth kinetics are related to the applied potential and added additive. The work is self-supportive and interesting for the mechanism understanding of the aqueous electrochemical synthesis.

However, to what extent the influence of the applied potential has on the porosity is not clear. Porosity should be calculated and given values.

Answer: The porosity has been calculated for all 10 sets of samples (aqueous, non-aqueous, without additives, with additives). Values are listed in Tab. 1 of revised manuscript.

We also included the following clarification:

“To create porous electrode, the co-generation of hydrogen along with metal deposition, represents a scientific interest for long time¹⁻³. This co-generation of hydrogen during the platinum ion reduction takes place at the electrochemical generation of platinum black at the cathode in aqueous media¹⁻³ or in non-aqueous media⁷. The role of hydrogen abundance at the cathode during the metal deposition is directly connected with the porosity of the metal foam formation. There is a recent review which compile the factors of influence (electrolyte composition, temperature, applied potential) at the formation of foam using the hydrogen bubble templating method¹⁸. Aligned to this idea, that the abundance of hydrogen increases at the more negative potential (i.e. from -0.7V to -1V), we observed an accentuated dependence of the porosity on the applied potential with all other electrolysis conditions being identical: at -0.8 V, the crystallite size was 20 nm; at -0.9 V, it became 35 nm; at -1 V, it was 50 nm (Fig. 5). The pore sizes of the sponge platinum in aqueous electrochemistry are 100 times larger and can reach 300 nm (Figs. 6). One can consider two levels of porosities: 1) the pores inside the porous platinum structure (Fig. 4) and 2) the pores between the porous platinum structures (Figs. 3). We stated in the Table 1 that the porosity of platinum layer increases in both aqueous and non-aqueous media at the more negative potentials from -0.7V to -1.5V. A plausible explanation to this phenomenon can be that the more abundant hydrogen evolution at more negative potential increases the entropy of the system and the structures are less compact.”

No practical controlment or application of the porosity based on the experiment interpretation is demonstrated in this work.

Answer:

Targeted application of porous Pt as broadband absorber has been added to the introduction together with addition of recent works discussing the application of porous Pt as a broadband absorber.

”Application of porous platinum layers

Electrosynthesized porous platinum is a perfect broadband absorber which is stable at high temperatures. Therefore electrosynthesized porous platinum is used as an optical absorber material in high temperature thermosensors.

By adjusting thicknesses and porosity of the layer the optical absorbance can be fine-tuned and make electrosynthesized porous platinum also interesting for plasmonics.

The large surface-area-to-volume ratio of highly porous platinum nanolayers determine their catalytic and electrocatalytic properties and is of advantage for implementation in fuel-cells, chemical, petrochemical, and pharmaceutical industry.

Furthermore, electrodes coated with porous platinum nanolayers can be used in high resolution electrochemical sensing due to the efficient electric current exchange and excellent structural stability of porous platinum¹⁹.

Due to the large chemical and thermal stability of porous platinum nanolayers with well-controlled thickness and porosity, this material is advantageous for use as shelter-protection of metals with high chemical reactivity in pyro-techniques.

The water-free fabrication of electrosynthesized porous platinum in combination with the possibility to grow it highly localized on 2D or 3D substrates opens novel nanotechnology fabrication routes in optoelectronics.“

Discussion on the results from potentiometric measurements has been revised. The experimental results from potentiometric measurements have been used to control porosity of porous Pt (table 1).

Besides, FFT for TEM images in Figure 1 is not indexed. Figures 6 and 7 are referred to in the main text in the reversed order. Figure 7 is mislabeled.

Answer: The FFT for HRTEM images are not indexed because the HRTEM confirm the crystallinity without a clear documentation of crystal phase orientation. The XRD diagram provides the confirmation of the face centered cube with unit lattice of $a=3.9231\text{\AA}$.

The figures were reorganized. Order of numbers of the Figures 6 and 7 in the manuscript has been corrected. The former figure 7, now figure 10, was re-labeled.

Reviewer #3 (Remarks to the Author):

The paper investigated the deposition of Pt porous layers and the underlying growth mechanism and kinetics. The result is interesting and useful. I recommend it for consideration in Communication Chemistry after minor revision. Additional comments list as follows:

(1) From XRD and HRTEM image in Fig. 1D and 2E, why is the crystallinity of Pt layer so poor, as we known, electrodeposition is an effective method to produce high-quality films?

Answer:

HRTEM (Fig. 1D): In the old manuscript we used a Cu grid with initiation porous Pt layer for HRTEM imaging. However, the quality of HRTEM images was poor. This might be related with the dependence of the porosity of electrodeposited porous Pt layers on the material of underlying electrode and on the applied electric potential being necessary for electrodeposition. → In revised paper, we prepared porous Pt layers on a Pt coated grid. Now, as expected, the porosity of electrodeposited porous Pt layers on Pt coated grids and on high-quality Pt layers is comparable, e.g. SEM images.

XRD (Fig. 2E): In this work, we investigated the initiation of porous Pt layers and not high-quality Pt layers which have been prepared by electrodeposition. The porosity of the porous Pt layers has been analyzed with XRD (Fig. 1D). Interestingly, the porous Pt layers are still crystalline. This is demonstrated by the XRD.

(2) From FFT in Fig. 2E, the paper demonstrated the Pt nanoparticles is single crystalline. More data should be provided.

Answer: In the revised manuscript additional HRTEM and FFT data from porous Pt layers on a Pt coated grid are provided to demonstrate the crystallinity of porous Pt layers. We thank the reviewer for stressing this point.

HRTEM were recorded at the margins of the porous layer growth (Figure for Reviewer 3) on platinum coated copper grid. In the range larger than 500 nm some of the particles have facets (triangular or hexagonal). They appear to be single crystals fitting to the facets. The assemblies exhibit irregular shapes (Fig. 2D). In the range < 100 nm, the structures appear to be agglomerates of crystalline particles. In the range < 10 nm, single crystalline particles are observed. TEM images recorded on mature porous platinum layer show several types of particles at the margins of the layer, some of them are amorphous and some are crystalline in the core with an amorphous shell. The shape of the amorphous particles tends to be round, crystalline ones tend to be elongated. It might be possible that amorphous and crystalline particles form simultaneously.

Figure for Reviewer 3. Porous platinum layer seen in transmission electron microscopy. A-N, TEM images of the porous platinum growth on a platinum coated copper grid (A,B,C,D,F,H,M) and the Fast Fourier Transform, FFT, (E,G,J,K,L,N) of each indicated areas (I,II,III,IV,V,VI).

(3) The size of Pt nanoparticles from SEM (Fig. 1B) and HRTEM (Fig. 1E) is not matched.

Answer: The SEM and HRTEM shows porous platinum at different stages of growth. There is an initiation layer of Pt layer, which was shown in the HRTEM (old Fig. 1E) and a 100 nm thick porous Pt layer, which was shown in SEM (old Fig. 1B and 1C). We revised discussion of corresponding Figures and explained differences in porosity of initiation layer and of thick Pt layer as follows.

“TEM images recorded on mature porous platinum layer show several types of particles at the margins of the layer, some of them are amorphous and some are crystalline in the core with an amorphous shell. The form of the amorphous particles tends to be round, crystalline ones tend to be elongated. These shapes are similar to the ones observed in SEM images (Figs. 1B,C), in which the amorphous parts alternates with crystalline ones (Figs. 3D-E).”

REVIEWERS' COMMENTS:

Reviewer #1 (Remarks to the Author):

The authors have made extensive edits to their manuscript. New text has been added along with a restructuring of their figures. Much of the text has been improved and strengthened. My remaining concerns are for addressing further clarity of their achievements and accuracy in their reporting.

As a note, many of the figures referenced in their letter did not match the numbering in the revised text. This led to extra confusion initially and additional time required for a careful review. The authors should be consistent between the figures referenced in their letters and the revised text -- not referencing old figure numbers that we no longer see in the revised text.

1) In Figure 12, the terms DL and ZDL need to be defined in the caption and text. The authors should also use consistent labelling for their notations.

2) The units were clarified in the methods section that relate to the reported percentages of the additives and reagents, but these units (e.g., g/g) should be added into the text and figure captions upon first use for clarity to the reader and this context is needed to interpret the data.

3) The authors state that they observed 25 and 35 nm pores in the HRTEM data in Figure 2, but this scale of nanopores is not observed therein. The authors need to include data that sufficiently supports this claim or revise their statements to be accurate to the data that is presented.

4) The authors state that the cubic lattice of the Pt observed by XRD correlated to the cubic morphology they present in their schematics. The cubic structure of the atomic lattice does not always correlate with the nanoscale structure of a material. The authors should revise their schematics to be more accurate to their observed results.

5) The authors state that the Pb is observed in the EDX at 0.3% Pb, but it appears that they also observe it at 0.01% Pb additives as well. They should reexamine the data; possibly I am wrong, but they seem to indicate and label a peak accordingly. It would have been better to also include techniques that can more accurately discern Pt from Pb at lower levels, such as through XPS analysis.

6) The authors state that their observed IR absorbance is "in agreement with our previous supported data", but in fact their material presented here still appears to be worse than what they previously reported. This deserves a careful review and discussion by the authors as to why the level of absorbance is less for these materials in contrast to their prior work.

7) The authors also state that "By adjusting thicknesses and porosity of the layer the optical absorbance can be fine-tuned and make electrosynthesized porous platinum also interesting for plasmonics." They report a thickness and estimated pore size of many materials prepared over a 5 year period. They should definitively demonstrate that they have the ability to control the thickness with a few supporting analyses, and more importantly that they can achieve the stated pore sizes.

Reviewer #2 (Remarks to the Author):

No further comments.

Reviewer #3 (Remarks to the Author):

The paper was well revised and I recommend it for acceptance in Communications Chemistry.

To the Editor

Manuscript: COMMSCHEM-20-0449A

We thank you and reviewers for the constructive criticism. Hereby listed is the point-by-point answer to the Reviewer questions:

Reviewer #1 (Remarks to the Author):

The authors have made extensive edits to their manuscript. New text has been added along with a restructuring of their figures. Much of the text has been improved and strengthened. My remaining concerns are for addressing further clarity of their achievements and accuracy in their reporting.

As a note, many of the figures referenced in their letter did not match the numbering in the revised text. This led to extra confusion initially and additional time required for a careful review. The authors should be consistent between the figures referenced in their letters and the revised text -- not referencing old figure numbers that we no longer see in the revised text.

Answer: We are thankful to the reviewer-1 for meticulously reviewing our manuscript. We are also grateful for your constructive remarks, which help us improve our manuscript. We have revised our manuscript accordingly.

1) In Figure 12, the terms DL and ZDL need to be defined in the caption and text. The authors should also use consistent labelling for their notations.

Answer question 1) Thank you for this remark. The error was fixed and the manuscript was checked for the consistence of notations.

2) The units were clarified in the methods section that relate to the reported percentages of the additives and reagents, but these units (e.g., g/g) should be added into the text and figure captions upon first use for clarity to the reader and this context is needed to interpret the data.

Answer question 2) The units of the involved percentages of the additives and reagents were added into the text and figure captions at their first apparition in the manuscript.

3) The authors state that they observed 25 and 35 nm pores in the HRTEM data in Figure 2, but this scale of nanopores is not observed therein. The authors need to include data that sufficiently supports this claim or revise their statements to be accurate to the data that is presented.

Answer question 3) We revised the statement relating to the porosity and HRTEM data in the Figure 2.

4) The authors state that the cubic lattice of the Pt observed by XRD correlated to the cubic morphology they present in their schematics. The cubic structure of the atomic lattice does not always correlate with the nanoscale structure of a material. The authors should revise their schematics to be more accurate to their observed results.

Answer question 4) The statement relating to the figure 1 was accordingly changed. From XRD analysis, the fcc lattice constant is 3.9231Å, which fits to the database value of the metallic platinum fcc lattice constant. From here we assume that the nanoscale crystallite contains platinum fcc unit

cells. The absence of hydrogen and additive during the electroreduction of platinum ion causes the formation of fcc metallic platinum. However, the formation of the metallic layer is interrupted in the presence of hydrogen and additives. The metallic crystals remain in the structure at different orientation, sizes and positions creating disorder and porosity.

5) The authors state that the Pb is observed in the EDX at 0.3% Pb, but it appears that they also observe it at 0.01% Pb additives as well. They should reexamine the data; possibly I am wrong, but they seem to indicate and label a peak accordingly. It would have been better to also include techniques that can more accurately discern Pt from Pb at lower levels, such as through XPS analysis. Answer question 5) The text was changed as follows:

"EDX does not show the presence of lead in the porous platinum layer obtained in non-aqueous media, only a weak Pb-La emission at 10.6 keV. However, the presence of lead with the lines Pb-La,b in platinum porous layer obtained in aqueous media is indicated by EDX (Fig. 5G,H) and SEM (Fig. 5C,D). The EDX peak of Cu-L α , β impurity derives from the electrode contacts (Fig. 3A, Fig. 5E-F). Lead is removed from the porous platinum layer by dissolution with HClO₄ or by sublimation at 600°C³. "

The interest in our working group thermosensorics lies rather on IR absorbance than on fine composition analysis of the broad band absorber material. XPS analysis of the broad band absorber material will be the subject of further investigations.

6) The authors state that their observed IR absorbance is "in agreement with our previous supported data", but in fact their material presented here still appears to be worse than what they previously reported. This deserves a careful review and discussion by the authors as to why the level of absorbance is less for these materials in contrast to their prior work.

Answer question 6) In our previous work we focused on fabrication of nanoporous platinum layers on many different electrodes. The IR absorbance has been shown to be best for nanoporous platinum layers on copper and silver electrodes. In this work we studied the mechanism of formation of nanoporous Pt layers and excluded the influence of electrode material on the process. Therefore, in order to exclude secondary effects from less noble metals during the electrodeposition, we used Pt anode, Pt cathode, Pt reference electrode. For sure the smaller conductivity of platinum (Pt: $9.3 \cdot 10^6$ A/V, Cu: $58.7 \cdot 10^6$ A/V, Ag: $62.1 \cdot 10^6$ A/V) and the achieved level of absorbance of nanoporous Pt layers on platinum is smaller than the level of absorbance of nanoporous platinum on copper and on silver.

The structure of porous platinum is mainly a function of the electric conductivity of the substrate, morphology of the substrate, electrolyte composition, operating temperatures, and applied potential.

In the revised manuscript we added a short explanation why and how we excluded the influence of electrode material on the electrodeposition and how the formation of nanoporous Pt layers depends on the conductivity of the substrate.

"The influence of the substrate conductivity upon the nanoporous platinum layers. The infra-red absorbance has been shown to be best for nanoporous platinum layers on copper and silver electrodes⁷. In this work we studied the mechanism of formation of nanoporous platinum layers and excluded the influence of electrode material on the process. We used platinum anode, platinum cathode, platinum reference electrode to exclude secondary effects from less noble metals during the electrodeposition. The structure and morphology of nanoporous platinum layers are determined by the electric current density on the cathode, which is a direct function of the electric conductivity of the cathode material. To achieve the similar structure of the porous platinum on different substrates made of flat metallic nanolayers by electrolysis of PtCl₄, one has to consider the electric conductivity of the substrate. Platinum has a conductivity of $9.3 \cdot 10^6$ A/V and silver of $62.1 \cdot 10^6$ A/V²².

This means that under identical electrolytic conditions (cathode area and roughness, applied potential, electrolyte concentration, temperature, electrode position) the time needed to build porous platinum of same thickness on platinum and on silver is by a factor of 6.7 longer for the platinum substrate in comparison to the silver substrate ($62.1 \cdot 10^6 \text{ A/V} / 9.3 \cdot 10^6 \text{ A/V} = 6.7$). Namely, porous platinum of similar thickness would be obtained if it grows for 60 s on platinum substrate and for 9.5 s on silver ($60 \text{ s} / 6.7 = 9 \text{ s}$). Using the same algorithm of calculation and assuming identical electrolytic conditions, to grow porous platinum of similar thickness for platinum with $9.3 \cdot 10^6 \text{ A/V}$ conductivity it requires 60 s, for copper with $58.7 \cdot 10^6 \text{ A/V}$ conductivity it requires 9.5 s, for gold with $44.2 \cdot 10^6 \text{ A/V}$ conductivity it requires 12.5 s, for aluminium with $36.9 \cdot 10^6 \text{ A/V}$ conductivity it requires 15.0 s, for molybdenum with $18.7 \cdot 10^6 \text{ A/V}$ conductivity it requires 30.0 s, for zinc with $16.6 \cdot 10^6 \text{ A/V}$ conductivity it requires 33.3 s, for nickel with $14.3 \cdot 10^6 \text{ A/V}$ conductivity it requires 40 s, for palladium with $9.5 \cdot 10^6 \text{ A/V}$ conductivity it requires 59 s, for steel with $10.1 \cdot 10^6 \text{ A/V}$ conductivity it requires 54.5 s, for tin with $8.7 \cdot 10^6 \text{ A/V}$ conductivity it requires 64 s, for lead with $4.7 \cdot 10^6 \text{ A/V}$ conductivity it requires 120 s, for titanium with $2.4 \cdot 10^6 \text{ A/V}$ conductivity it requires 240 s, for stainless steel with $1.28 \div 1.37 \cdot 10^6 \text{ A/V}$ conductivity it requires $408 \text{ s} \div 438 \text{ s}$, for FeCrAl with $0.74 \cdot 10^6 \text{ A/V}$ conductivity it requires 750 s."

Electric conductivity source: https://www.tibtech.com/conductivite.php?lang=en_US
(8.04.2021_11:05)

7) The authors also state that "By adjusting thicknesses and porosity of the layer the optical absorbance can be fine-tuned and make electrosynthesized porous platinum also interesting for plasmonics." They report a thickness and estimated pore size of many materials prepared over a 5 year period. They should definitively demonstrate that they have the ability to control the thickness with a few supporting analyses, and more importantly that they can achieve the stated pore sizes.

Answer question 7) As an example, in this work we demonstrated how the structure of the nanoporous Pt layers can be controlled by the potential and by the electrolyte composition (aqueous and non-aqueous media). For a given potential and electrolyte composition the thickness is well controlled by the electrodeposition time. In Tab. 1 we list results from the thickness control using potential, electrolyte composition and electrodeposition time. To analyze the layer thickness and porosity we used Focus Ion Beam (FIB) cut and SEM imaging of the layer transversal cut or scratches. We stated in Table 1 that the porosity of platinum layer increases in both aqueous and non-aqueous media at the more negative potentials from -0.7V to -1.1V. A plausible explanation to this phenomenon can be that the more abundant hydrogen evolution at more negative potential increases the entropy of the system and the structures are less compact. The layer thickness increases with the time of the electrolysis in both aqueous and non-aqueous media.

There are studies in our lab relating to the application of porous platinum for plasmonics, however these will constitute the subject of another article.

Reviewer #2 (Remarks to the Author):

No further comments.

Answer: Thank you.

Reviewer #3 (Remarks to the Author):

The paper was well revised and I recommend it for acceptance in Communications Chemistry.

Answer: Thank you for your recommendation for publication.

REVIEWERS' COMMENTS:

Reviewer #1 (Remarks to the Author):

The authors have made all the necessary corrections and additions. I believe that this revised work is ready for acceptance for publication.